# Salicylic acid metabolism and signalling coordinate senescence initiation in aspen in nature

Jenna Lihavainen [1], Jan Šimura[2], Pushan Bag [1,4], Nazeer Fataftah[1], Kathryn Megan Robinson [1], Nicolas Delhomme [2], Ondřej Novák[2,3], Karin Ljung [2] & Stefan Jansson [1] ✉

Deciduous trees exhibit a spectacular phenomenon of autumn senescence driven by the seasonality of their growth environment, yet there is no consensus which external or internal cues trigger it. Senescence starts at different times in European aspen (*Populus tremula* L.) genotypes grown in same location. By integrating omics studies, we demonstrate that aspen genotypes utilize similar transcriptional cascades and metabolic cues to initiate senescence, but at different times during autumn. The timing of autumn senescence initiation appeared to be controlled by two consecutive "switches"; 1) first the environmental variation induced the rewiring of the transcriptional network, stress signalling pathways and metabolic perturbations and 2) the start of senescence process was defined by the ability of the genotype to activate and sustain stress tolerance mechanisms mediated by salicylic acid. We propose that salicylic acid represses the onset of leaf senescence in stressful natural conditions, rather than promoting it as often observed in annual plants.

Deciduous trees in temperate regions salvage nutrients through senescence before abscission of the leaves in autumn, which has massive ecological importance. Although beautiful autumn colours are widely appreciated by the public, senescence regulation at the molecular level is not well understood. Research efforts[1–3], such as genetic approaches, transcriptomics, metabolomics and external applications of phytohormones have identified genes where mutations lead to premature or delayed senescence, senescence-associated genes (SAGs) that are up- or downregulated in senescing leaves, and metabolic signals such as reactive oxygen species (ROS) that are involved in the process[4–8]. Yet, there is no consensus on the senescence trigger, which metabolites, genes or post-translational mechanisms are the most important ones—across species—for the process, or which environmental factors could consistently explain why senescence starts at a certain time in trees in nature.

In our previous studies of autumn senescence in European aspen (*Populus tremula* L.) we found that a given aspen genotype in its natural environment typically initiates senescence (determined by rapid chlorophyll depletion) on around the same date every year[9,10]. However, the date shifts if the same genotype is instead grown either in a greenhouse[10], or at another latitude[11], if the tree is girdled[12] or fertilised[13]. Moreover, with multi-year studies of one genotype, we previously proposed that the main variation represented by a slow and gradual shift in the transcriptome profile[14] or cytokinin levels[15] can be difficult to link with senescence onset under natural conditions. These observations suggest that senescence in deciduous trees cannot be simply explained by looking at the global transcriptome pattern or by a single hormone signalling pathway; rather it is most likely affected by an intricate regulatory network that integrates both external environmental cues and internal metabolic signals.

[1]Umeå Plant Science Centre, Department of Plant Physiology, Umeå University, 90189 Umeå, Sweden. [2]Umeå Plant Science Centre, Department of Forest Genetics and Plant Physiology, Swedish University of Agricultural Sciences, 901 83 Umeå, Sweden. [3]Laboratory of Growth Regulators, Faculty of Science, Palacký University and Institute of Experimental Botany of the Czech Academy of Sciences, Šlechtitelů 27, CZ-783 71 Olomouc, Czech Republic. [4]Present address: Section of Molecular Plant Biology, Department of Biology, University of Oxford, Oxford, UK. ✉e-mail: stefan.jansson@umu.se

On one hand, a problem with our approach of studying senescence in natural conditions may be that many factors vary at the same time, which makes it difficult to separate the "true signal" from the environmental variation. On the other hand, our approach can have several benefits compared to experiments performed under controlled conditions, as the latter cannot separate a developmental programme from a timetable[14], that mature trees–that cannot be easily grown under controlled conditions–have more uniform senescence of the canopy than young plants that typically show progressive senescence starting from the oldest bottom leaves, and that trees are likely to integrate multiple signals to know that it is autumn. Therefore, by taking advantage of our previous studies of autumn phenology in populations of natural aspen accessions sampled from a large latitudinal range[10,11], we aimed to understand the cellular programme leading to senescence onset in several genetically different aspen trees that vary substantially in their senescence onset dates in a common garden. Theoretically, the changes that truly regulate the onset of autumn senescence should precede or coincide with its onset, initially in the genotype that starts to senesce early and later in the genotype that senesces late in autumn, assuming that they utilise similar transcriptional, hormonal and metabolic programmes to integrate the signals to initiate senescence.

Hence, we have integrated transcriptomics and metabolomics with co-expression network analyses to understand why aspen genotypes start to senescence at different times, although grown in same location. We demonstrate that the trees integrate many environmental and internal signals through an intricate regulatory network to properly time their senescence onset. The information is transduced via – at least – two parallel cascades that are interlinked with the salicylic acid (SA) signalling pathway that is repressed at senescence onset along with decreased levels of SA and increased levels of its catabolite 2,3-dihydroxybenzoic acid. Our study demonstrates that aspen genotypes display similar transduction pathways to initiate senescence, but at different times during autumn and that SA represses autumnal leaf senescence onset in aspen in natural conditions by promoting defence mechanisms, rather than advancing it as often observed in annual species[5,16–19].

## Results

### Senescence phenotypes are consistent in the field and in the greenhouse

We have studied the variation in autumn phenology and senescence onset dates in genotypes from the Swedish aspen collection (SwAsp) in a common garden and in a genotype 201 growing at the Umeå University campus (part of the Umeå aspen collection - UmAsp)[9–11,20]. For this study, we selected 201 and five SwAsp genotypes originating from southern (1, 33), middle (48) and northern (81, 96) latitudes of Sweden that showed large variation in senescence onset dates (Fig. 1a, see the framework of analyses in Fig. S1). In 2011, 201 started to senesce around 255 DOY (the day of the year) within the typical range for this genotype[9,10] (Fig. 1) and its clonal replicates in the common garden showed similar onset date to the parent tree in 2018 (Fig. 1c and Fig. S2), despite the different weather conditions in the two study years (Fig. 1d, e and Table S1). In 2018, 81 started to senescence around 242 DOY, 96 on 244 DOY, 48 on 247 DOY, 33 on 263 DOY and 1 on 264 DOY close to the mean senescence date over the two study years (Fig. 1b, c). Based on the senescence timing in the aspen population (Fig. S2), these genotypes were classified as early- (around or before 245 DOY, hereafter E81, E96), intermediate- (between 245 and 260 DOY, hereafter I48 and I201) and late-senescence (after 260 DOY, hereafter L1, L33) phenotypes (Fig. 1). Although the timing of senescence onset was later, the variation between the three genotypes selected for the transcriptome profiling (L1, I48, E81) persisted in the greenhouse[10] under conditions with constant air temperature (C°) and relative humidity (RH%) and with natural light

and photoperiod (Fig. S2). This indicated that the variation in air temperature in the field could advance senescence onset, but also that other factors–genetic background, other environmental variation and/or metabolic status–would account considerably for the differences in senescence timing.

### The shift in the global transcriptome profile does not reflect the variation in senescence onset

To understand why the genotypes start to senescence at different times in autumn in the common garden, we compared transcriptome profiles of one each early- (E81), intermediate- (I48) and late-senescence genotypes (L1) during autumn 2018 and complemented the analyses with transcriptome data from I201 from year 2011. In both years, the first principal components (PCs) that explain most of the variation in the data displayed gradual shifts of the global transcriptome profile through autumn (Fig. 2a, b, Fig. S3). However, it did not reflect the large difference in senescence timing between the SwAsp genotypes (23 days between E81 and L1, Fig. 2a, b). Thus, the variation in senescence onset could not be explained by looking at the main temporal shift in the global transcriptome profile.

We then compared the lists of differentially expressed (DE) genes in SwAsp genotypes to identify similarly and differentially regulated genes during autumn (DESeq2, Wald test, FDR adjusted $P < 0.01$ two-sided, Fig. S4, Supplementary Data 1). In line with the variation seen in PCA, 511 and 353 genes were similarly up- or downregulated, respectively, during autumn irrespective of genotype, and over the two years (Fig. 2d), and those genes are typically regarded as senescence-associated genes (SAGs). We compared our list of genes with those obtained from two transcriptomics studies of poplar (Fig. 2e) (*Populus trichocarpa*)[21,22] and found consistently upregulated transcription factors (TFs) during autumn in both species related to innate immunity, stress and defence responses such as *WRKY75, WRKY48, NAC100, NAC072* and *TGA1* and the repressed genes enriched with gene ontology (GO) terms related to chloroplast processes (Fig. 2e, f, Supplementary Data 2). However, the expression of individual senescence-enhanced TFs could not explain the variation in senescence onset as they displayed similar temporal patterns in the SwAsp genotypes (Fig. 2g).

Therefore, we performed weighted gene co-expression network analysis (WGCNA, details in Supplementary Materials)[23,24] that considers correlation between genes in individual genotypes as well as in a consensus network (results in Figs. S5–S15 and Supplementary Data 3–6). With this approach, we could compare the transcriptome patterns among the genotypes in terms of gene modules and find significant correlations (Pearson $r$, $P < 0.05$ two-sided) of the signature expression patterns of the modules (eigengenes) and individual genes with weather parameters and metabolite levels (see result in Figs. S5–S15 and Supplementary Data 7–13) to identify the sources underlaying the transcriptome responses.

In both years, module network structures displayed broadly two main transcriptional cascades changing in parallel in autumn (Fig. 3a, Figs. S11 and S13). The largest modules contained SAGs, chromatin remodelling factors (module V) and displayed gradually enhanced or repressed expression during autumn (cascade-1, Fig. 3a, b). However, the modules in cascade-1 displayed similar temporal patterns in three SwAsp genotypes that could not explain senescence onset (Fig. 3b). On the other hand, the expression patterns in cascade-1 followed the decrease in air temperature and the levels of cytokinin (CK) and auxin (IAA) metabolites (Fig. 3a, Figs. S10 and S11). As a general pattern, the levels of active free forms of CKs and IAA decreased during autumn, whereas the levels of their conjugated forms or catabolites increased, and the timing was similar in the SwAsp genotypes (Fig. 4a, b, Fig. S9). Furthermore, module patterns in cascade-1 and genes encoding enzymes participating in chlorophyll metabolism correlated with chlorophyll levels in the genotypes that senesced in early (E81) or mid-

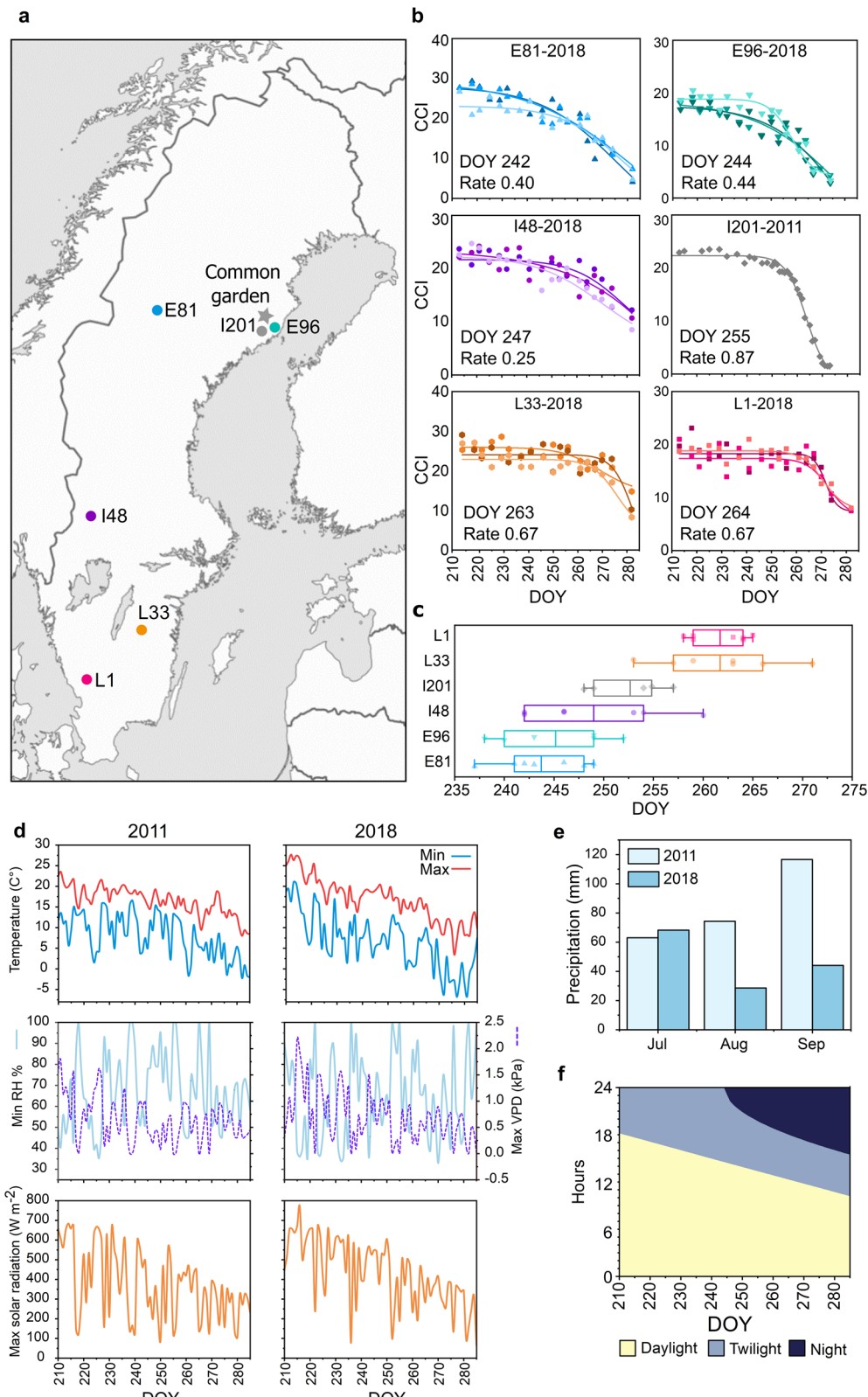

autumn (I48, I201), but not in the late one (L1) that showed changes in gene expression much earlier than when its chlorophyll content started to decrease (Figs. S10, S14 and S15, Supplementary Data 13). Evidently, the gradual shift in transcriptome and CK and IAA metabolite levels did not correlate with the onset of senescence across genotypes or study years, but they could advance senescence in response to decreasing air temperature after it was initiated.

## Activation of stress signalling cascades precedes autumn senescence

Above and beyond the gradual changes in the transcriptome, we detected another cascade that displayed pronounced and often transient changes in expression patterns during autumn (Fig. 3a, b). In 2018, the transcriptome responses on 237 and 253 DOY seemingly coincided with cloudy conditions that persisted over a few days

**Fig. 1 | Autumn senescence phenotypes.** Swedish aspen (SwAsp) genotypes originating from the south (L1, L33), central (I48) and north (E81, E96) of Sweden were grown in a common garden in Sävar near Umeå (marked with a star, **a**). Genotype I201 is local to Umeå, situated at Umeå University campus, and its clones are part of the Umeå aspen collection grown in a common garden in Sävar (**a**). The onset and the rate of autumn senescence were determined based on the chlorophyll content index (CCI). Chlorophyll curves (CCI values as a mean of five leaves) are shown for the campus tree, I201, in autumn 2011 and for the three replicate trees of SwAsp genotypes in autumn 2018 (**b**). The variation in senescence onset date over the two study years is shown in a box plot representing mean (solid line), 25 and 75%

quartiles and minimum and maximum values (whiskers) (**c**). Points present data over the two study years, data are from one parent tree in 2011 and four clonal replicates in the UmAsp collection in 2018 of genotype I201 ($n = 5$), and from three to four replicates of the SwAsp genotypes in each study year ($n = 7$, except $n = 6$ in E96). Minimum and maximum air temperature (C°), minimum relative air humidity (RH %), maximum vapour-pressure deficit (VPD, kPa) and maximum solar radiation (W m$^{-2}$) in autumn 2011 and 2018 (**d**) in Umeå. Precipitation (monthly sum, mm) in autumn 2011 and 2018 (**e**) and the seasonal shift in the light environment in Umeå (**f**). Source data are provided as Source Data files.

(Figs. 1c, 3a and Fig. S8). In addition to low solar radiation and elevated air humidity (high RH% and low vapour-pressure deficit, VPD, Fig. 1c), the overcast days are accompanied by increased blue-to-red (B:R) and red-to-far-red (R:fR) ratios of the light spectra (Fig. S16). These short-term environmental fluctuations affected a vast number of genes acting at multiple regulatory levels—transcription (light signalling, circadian rhythm), post-transcription (RNA metabolism), chromatin (histones, chromosome organisation) and translation (ribosome, translational initiation, cascade-2 Fig. 3a, Fig. S11, Supplementary Data 4). Although most of the responses in cascade-2 were similar in the three SwAsp genotypes, the more intense upregulation of genes encoding cytoplasmic translational initiation factors on 237 DOY in E81 and I48 than in L1 (module XX) projected downstream to enhance expression of RAV (Related to ABI3-VP1) TFs (module XXVI hubs) and other genes involved in ethylene (ET) and abscisic acid (ABA) signalling (AP2/ERF TFs), abiotic stress responses, and programmed cell death (PCD, Figs. 3a, b, 5a, Fig. S17, Supplementary Data 1). These responses preceded senescence onset occurring earlier in genotypes with early- and intermediate-senescence and later in the season in the late-senescing genotype in response to similar weather shifts (Fig. 5a). Overall, the expression of positive PCD regulators were the highest in the genotype with the earliest senescence, increasing during autumn and amplified by the weather shifts (Fig. S17).

In line with the pronounced induction of genes encoding translation initiation factors and ribosomal proteins in early autumn, the endoplasmic reticulum (ER) stress- and unfolded protein response (UPR)-related gene module was upregulated in mid-autumn (241–246 DOY) in the SwAsp genotypes (module XVIII, Fig. 3a, b). Similarly, ER stress-related genes were upregulated in I201 during mid-autumn 2011 (modules 6, 13 and 16, Fig. S13). The start of the night period (when solar elevation angle drops more than 18° below the horizon) at this latitude on 245 DOY (2$^{nd}$ September, Fig. 1f) suggests that on top of the short-term fluctuations in light conditions, the seasonal shift in light regime could underly the transcriptional responses in cascade-2. In line with this, genes involved in light signalling and circadian rhythm were repressed in mid-autumn (Fig. 3a, b), and some of them even more in E81 and I48 than in L1 (Supplementary Data 1).

Overall, the transcriptional responses in cascade-2 reflected frequently occurring stress events (hereafter 237–253 DOY regarded as the *stress period*) that were accompanied by mainly similar metabolic perturbations such as the accumulation of auxin catabolite (oxIAA), several amino acids and tricarboxylic acid (TCA) cycle intermediates (Figs. 4b, d, 6d and Supplementary Data 14). Stronger metabolic perturbations in the early-senescing genotypes than in the late-senescing ones (Fig. 6b and Fig. S18) indicated that the genotypes differed in their ability to maintain cellular homoeostasis under frequently challenging autumn conditions. These findings were further corroborated in I201 in autumn 2011; where the metabolic perturbations occurred and abiotic stress-responsive genes and a gene module associated with cell death regulation were upregulated at the start of the night period before senescence onset (modules 11 and 16, Figs. S13, S18 and S19).

Evidently, autumn was accompanied by environmental variation that disrupted metabolic processes and triggered the re-wiring of

the transcriptional network inducing pro-senescence factors: cytoplasmic eukaryotic translation initiation factors (eIFs/TIFs), abiotic stress signalling (ET, ABA) and PCD genes, thus we defined this as the transition into a *pro-senescence phase*. The timing of these responses suggested that both short-term and seasonal shifts in the light regime could underly the observed responses. The genotypes that senesced early showed stronger responses earlier in the season than the ones that senesced later, indicating that the genotypes differed in their sensitivity to these fluctuations. Since the shift into a *pro-senescence phase* preceded the onset irrespective of genotype it appeared to predominantly predispose the trees for senescence.

## The repression of salicylic acid signalling pathway coincides with senescence onset

The two major transcriptional cascades that responded to gradual (cascade-1) and short-term (cascade-2) environmental variations were both in the end interlinked with modules that were first enhanced and then transiently repressed that coincided with the start of rapid chlorophyll depletion (*senescence phase*) in the SwAsp genotypes (modules IX and X, Fig. 3b). These modules were enriched with genes involved in the regulation of salicylic acid (SA) metabolism and signalling pathway including genes encoding leaf senescence, immune response and cell death regulators, several NAC and WRKY TFs and receptors (Fig. 3a, Supplementary Data 4). In line with this, the genes encoding membrane-bound proteins localised in the cell periphery at the interface of signal perception and within the ER showed significantly different dynamics among the three SwAsp genotypes (Fig. 2c, Supplementary Data 1). The hub genes encoded positive regulators of the SA metabolism and signalling pathway, systemic acquired resistance (SAR) and hypersensitive response (HR) such as two SARD1 (SAR DEFICIENT 1)-like TFs, calmodulin-binding protein CBP60B (CALMODULIN-BINDING PROTEIN 60B) and EDS1 (ENHANCED DISEASE SUSCEPTIBILITY1) (Supplementary Data 4). Similarly, in the I201 network, *SARD1* acted as an important hub gene, upregulated during autumn but transiently repressed at the onset of chlorophyll depletion (module 4, Fig. S13, Supplementary Data 6). These results suggested that the repression of the SA-mediated transcriptional programme after *pro-senescence phase* could initiate the senescence process.

We confirmed that SA levels correlated positively with the expression of *SARD1* and *CBP60B* involved in SA biosynthesis, and with other genes involved in SA signalling, evidently defining this transcriptional response in the SwAsp genotypes (Fig. 5b, c, Fig. S20, Supplementary Data 15). Furthermore, the levels of glycerol-3-phosphate (glycerol-3-P), considered a mobile signal inducing SAR[25,26], correlated positively with the expression of *SARD1* and *CBP60B* (Fig. S21, Supplementary Data 15). SA levels showed very different dynamics in the five SwAsp genotypes during autumn; the levels remained low in E81 with the earliest senescence onset, while they were induced in the intermediate- and late-senescing genotypes especially during the *stress period* in mid-autumn (Fig. 4c). SA is typically regarded to promote senescence, however in E96, a strong accumulation of SA in early autumn (>10 fold on 232 DOY) did not trigger senescence (Fig. 4c). On several occasions elevated SA levels were accompanied by

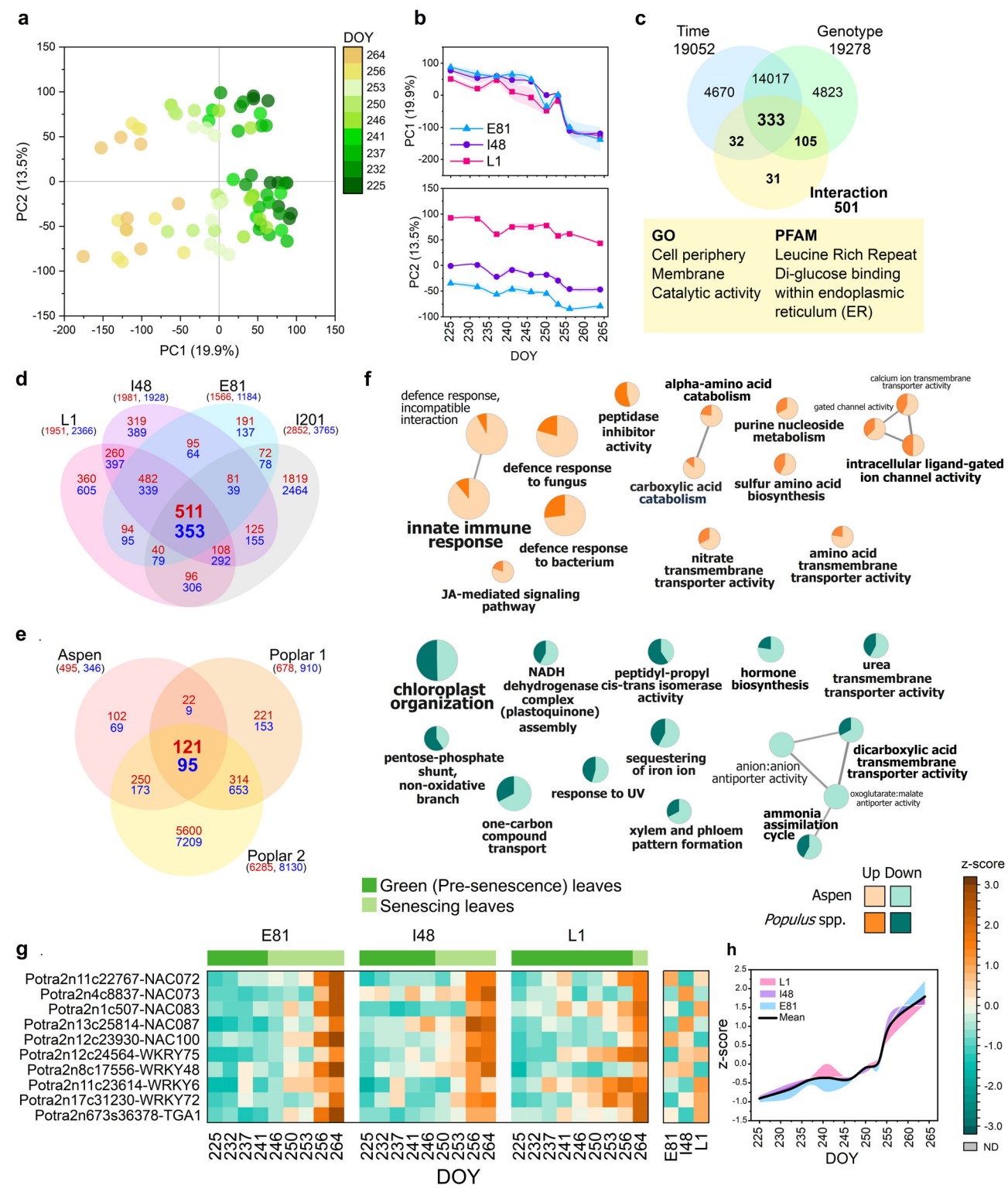

elevated levels of other stress-induced phytohormones, supporting that besides their absolute levels the antagonistic relationship between phytohormone signalling pathways determines the transcriptional response and physiological outcome (Fig. 4). Furthermore, SA levels correlated positively with the expression of *NCED3* encoding 9-cis-epoxycarotenoid dioxygenase, a key enzyme in the ABA biosynthesis, with *ACS1* encoding 1-aminocyclopropane-1-carboxylate synthase involved in ET biosynthesis, and with WRKY TFs involved in SA/JA (jasmonic acid) crosstalk highlighting the important role of SA in the phytohormone signalling network during autumn (Figs. S17, S20 and Supplementary Data 15).

The central role of SA in senescence regulation was further supported by the levels of the SA catabolite 2,3-dihydroxybenzoic acid (2,3-OH-BA) that increased during autumn earlier in early-senescing genotypes than in the ones that senesced later (Fig. 6c, d). Moreover, it was identified as a major hub in the metabolic network that displayed broadly two cascades like the transcriptional network. Metabolite levels showed gradual shifts during autumn (cascade-1, Fig. 6, Fig. S22)

**Fig. 2 | Global transcriptome patterns in three SwAsp genotypes in autumn 2018 and meta-analysis of up- and downregulated genes during autumn in *Populus* spp.** Principal Component Analysis (PCA) scores plot (**a**) and time-dependent plots of the major principal component (PC) scores (**b**) of leaf transcriptome in three SwAsp genotypes in autumn 2018. The main variation in the transcriptome was explained by time (225–264 DOY, the day of the year) during autumn (PC1) and based on genotype (PC2). Data are mean ± SE (shadowed area), $n = 3$ in each time point per genotype, except $n = 2$ in 237 DOY in genotype I48. Venn diagram depicts the number of genes with significant time, genotype and interaction (time × genotype) effects (**c**). The enriched Gene Ontology (GO) and PFAM (protein family and domain) terms for the gene set with a significant interaction effect determined based on Likelihood Ratio Test (LRT, **c**). Venn diagram displays the overlap of up- and downregulated ($\log_2$ fc > 1.0) genes in autumn in four genotypes of aspen (*P. tremula*, **d**), and in comparison with two data sets in

poplar (*P. trichocarpa*) (**e**). The lists of up- and downregulated genes in poplar leaves were obtained from Leaf Senescence Database[21](Poplar 1) and from Lu et al.[22] (Poplar 2). The number of upregulated genes is in red, downregulated in blue, and shared genes between the data sets in bold. The enriched GO terms for biological processes of consistently up- and downregulated genes during autumn in aspen and in *Populus* spp. (**f**). The GO term nodes are coloured based on the proportion of shared genes in aspen and *Populus* spp. The expression patterns of senescence-enhanced WRKY, NAC and TGA transcription factors during autumn in three SwAsp genotypes (**g**, **h**). The data in the heatmap are mean expression normalised to z-score and the line graph shows the average expression of heatmap genes during autumn in each genotype relative to the overall mean, coloured area presents the difference between the genotype mean and the overall mean. The list of SAGs and GO term enrichment results are in Supplementary Data 2. Source data are provided as Source Data files.

and abrupt perturbations (cascade-2, many of which showed significant time×genotype interactions, Fig. 6e). Slowly-developing senescence symptoms included an increase in branched chain (BCAA: Val, Ile, Leu) and other amino acids (Met, Phe), raffinose, α-tocopherol and orotic acid, and a decrease in Gly and four-carbon sugars that occurred earlier in E81 and E96 than in other genotypes (Fig. 6d, Fig. S22). Accordingly, the transcription of enzymes participating in raffinose biosynthesis and amino acid metabolism (BCAA biosynthesis and Ser/Cys metabolism) was upregulated earlier in E81 and I48 than in L1 (Fig. 6f). Since these symptoms occurred around the same time (E81, E96, I48, I201) or even before (L1, L33) the chlorophyll content started to rapidly decrease (Fig. S22), they were likely associated with the activated catabolism and nutrient remobilisation at senescence onset, and these processes may have started before the first visible symptoms appeared. Nevertheless, senescence onset estimated based on metabolic markers gave comparable dates than estimated based on chlorophyll content, also in another senescence experiments in aspen (Fig. S23, Supplementary Data 16).

Taken together, the upregulation of SA catabolism and the associated repression of SA levels and the SA-mediated transcriptional programme resulted in the start of metabolic senescence processes and the rapid chlorophyll depletion marking the senescence initiation in aspen. Considering all the presented data, we concluded that induced and sustained SA levels and the SA signalling pathway can repress senescence onset in aspen trees under stressful field conditions, rather than promote it as often reported in annual species under controlled conditions.

## Salicylic acid can alleviate proteotoxic stress to delay senescence onset

To understand how SA metabolism and SA-mediated signalling pathway could modulate senescence timing, a gene co-expression network was constructed with genes with consistent positive correlation with SA levels and their closest neighbours, that included RAV TFs and several SAGs (Fig. S17). Importantly, SA levels and SA-associated gene modules showed positive relationships with the ER stress-associated module and genes related to protein modification, glycoprotein and nucleotide sugar metabolism and ERAD pathway (ER-associated degradation of misfolded proteins by ubiquitination) with a stronger connection in L1 and I48 that induced SA levels in mid-autumn, and a weaker connection in E81 that did not (Figs. S10 and S20). Accordingly, like SA pathway genes, the expression of those genes was overall the lowest in E81 and strongly repressed in I48 in mid-autumn potentially predisposing them for senescence onset due to hindered proteostasis (Fig. 5a, Fig. S17, Supplementary Data 1). *BAG7* was an exception as it showed higher expression in E81 and I48 than in L1 in early autumn (Fig. 5a).

Moreover, network analysis identified Calreticulin 3 (*CRT3*) and other genes accounting for glycoprotein quality control in the ER (ERQC) by the chaperone system as consistent mediators controlling

the information flow through the network structure—between cascade-1, SA- and ER stress-associated gene modules (Fig. 5c, Fig. S17). Thereby, SA pathway and cascade-1 appeared to jointly mediate ER stress responses via the ERQC system that facilitates proper folding of glycoproteins, many of which are membrane-bound receptors. These findings were corroborated by the network analysis in I201 in 2011, where the two transcriptional cascades and the SA-associated gene module were connected to modules enriched with genes related to ER stress responses, glycosylation and cell death regulation directly upstream of the module containing genes encoding enzymes in metabolic pathways and receptor-like proteins (hubs) transiently repressed around senescence onset (Fig. S13, Supplementary Data 6).

## Sustained antioxidant defence delays senescence onset
Finally, as our network analyses revealed a positive relationship between SA signalling pathway, cellular respiration and redox status (co-expression with PNP-A, alternative oxidase [AOXs], nudix hydrolases [NUDT] and glutathione transferases [GST]) (Figs. 3a and 5c), we studied whether senescence onset was defined by the capacity of the different genotypes to induce enzymatic and metabolic antioxidant systems to counteract ROS and oxidative damage under constantly challenging autumn conditions. Indeed, the levels of lipid peroxidation marker, malondialdehyde (MDA), were the highest in the SwAsp genotype with the earliest senescence onset in 2018 (Fig. 7a, b) and a transient accumulation of MDA and hydrogen peroxide ($H_2O_2$) in mid-autumn preceded senescence onset in I201 in 2011 (Fig. S24). Despite the strong day-to-day variation, the early-senescing genotypes (E81, E96) had apparently low and less induction in superoxide dismutase (SOD) activity than the other SwAsp genotypes through autumn (Fig. 7c). They had also the lowest catalase (CAT) activity and strong fluctuations in $H_2O_2$ levels (Fig. 7b, d).

High metabolic hydroxy radical (˙OH) scavenging capacity was maintained in L1 throughout the autumn, whereas the early-senescing genotypes showed depleted and overall low ˙OH scavenging capacity (Fig. 7f) that could not be explained by the levels of ascorbate or glutathione that were high (Fig. 7a, Figs. S18 and S25). Instead, the lower total abundance of phenolic compounds in early-senescing genotypes than in the late-senescing ones could account for the observed differences in their metabolic ROS-scavenging capacity (Fig. 7e). This suggested that the metabolic constraints that predisposed the trees for early-senescence onset spanned further in the phenylpropanoid pathway beyond the SA biosynthesis branch, corroborated by higher levels of precursor metabolite shikimic acid, and by lower expression of genes encoding pathway enzymes in E81 than in L1 (Fig. 7e, g).

Taken together, the genotypes showed similar transduction pathways leading to senescence onset but at different times in autumn (Fig. 8). Evidently, autumn was characterised by shifts in environmental conditions that enhanced ROS formation, ER

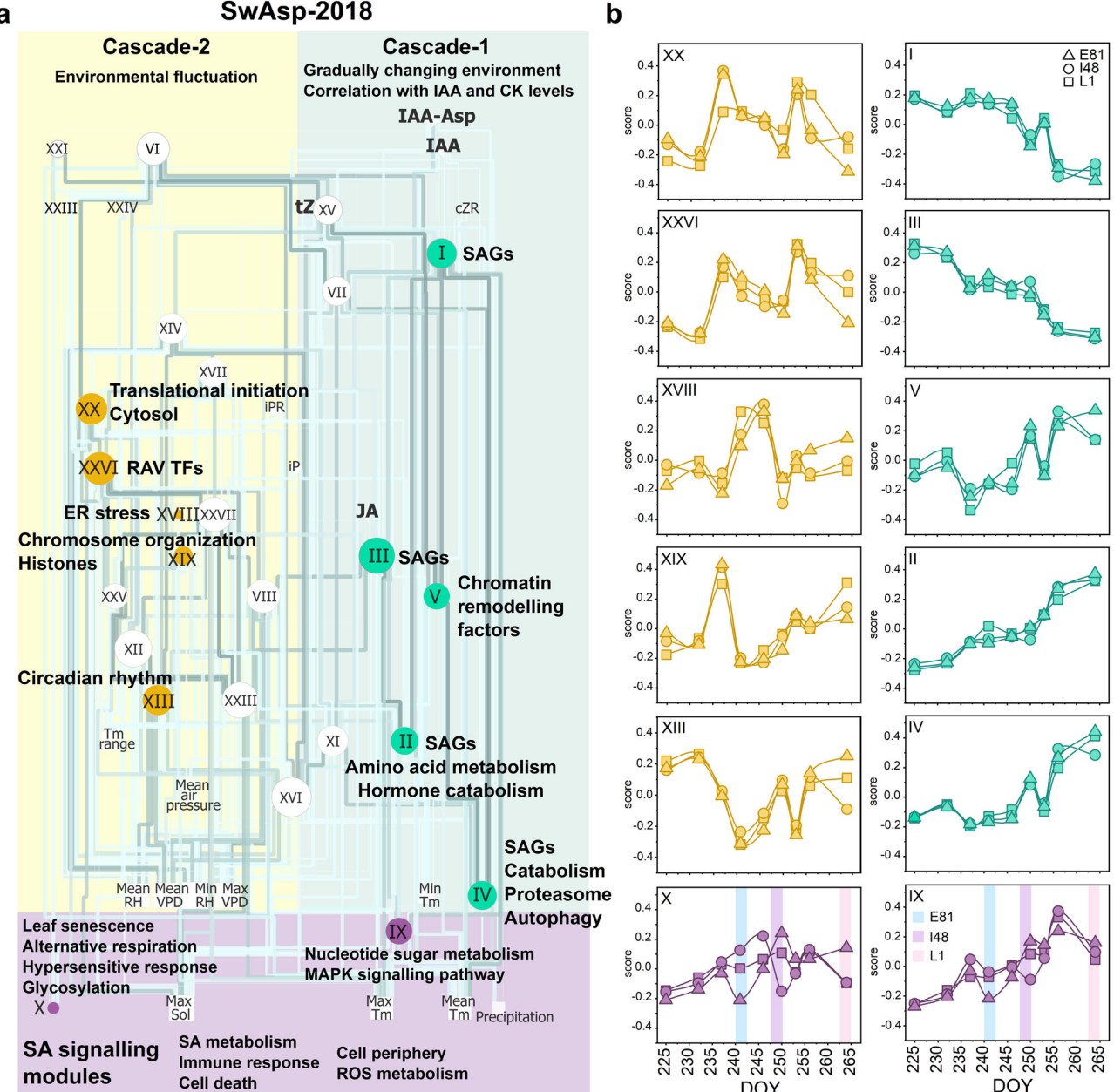

**Fig. 3 | Network analysis reveals two interlinked transcriptional regulatory cascades that respond to environmental cues and hormonal signals during autumn.** Weighted gene co-expression network analysis (WGCNA) was performed with transcriptome data from three aspen genotypes during autumn 2018. Network visualisation is based on the correlation between eigengenes (the signature expression pattern of the identified gene modules), weather parameters (past 24 h) and phytohormone levels (**a**) using a hierarchical layout in Cytoscape. Significant correlations (*P*-value < 0.05 two-sided in all genotypes) are depicted by edges that are coloured based on the correlation coefficient, darker shades depicting higher absolute Pearson *r* (**a**). The size of each node is proportional to its degree, larger node size depicting higher connectivity. Enriched Gene Ontology (GO) terms and Kyoto Encyclopedia of Genes and Genomes (KEGG) pathways in the gene modules are shown (**a**, see details in Supplementary Data 4). Cascade-1 comprises of modules that display gradual expression changes during autumn. Genes that are

repressed or induced during autumn are regarded as senescence-associated genes (SAGs, see details in Fig. 2 and Supplementary Data 1). Modules in cascade-2 show transient expression changes (**a**). The two cascades are interlinked and connected with two gene modules (IX, X) enriched with genes involved in salicylic acid (SA) metabolism and signalling (**a**). The patterns of selected eigengenes in three SwAsp genotypes (**b**), *n* = 3 in each time point per genotype, except *n* = 2 at 237 DOY in genotype I48. A shaded vertical line represents senescence onset in each genotype that coincide with the transient down-regulation of the two SA signalling modules (**b**). Modules are numbered based on the descending number of assigned genes and coloured based on the assigned cascade (**a**, **b**). See details and all eigengene patterns in Fig. S11. Source data are provided as Source Data files. ER endoplasmic reticulum, SAGs senescence-associated genes, tZ *trans*-zeatin, iP isopentenyl adenine, IAA indole-3-acetic acid, IAA-Asp Indole-3-acetic acid-aspartate.

---

stress and metabolic perturbations that could trigger senescence onset if appropriate defence mechanisms were not activated. Genotypes that did not induce and sustain SA signalling and antioxidant systems in response to stress signals senesced early, whereas the genotypes that actively induced and sustained them

senesced later (Fig. 8). Therefore, we propose a model (Fig. 9) where senescence onset in natural field conditions is defined by this ability of the different genotypes to counteract metabolic stress and pro-senescence signals that can be evoked and amplified by many environmental factors in autumn.

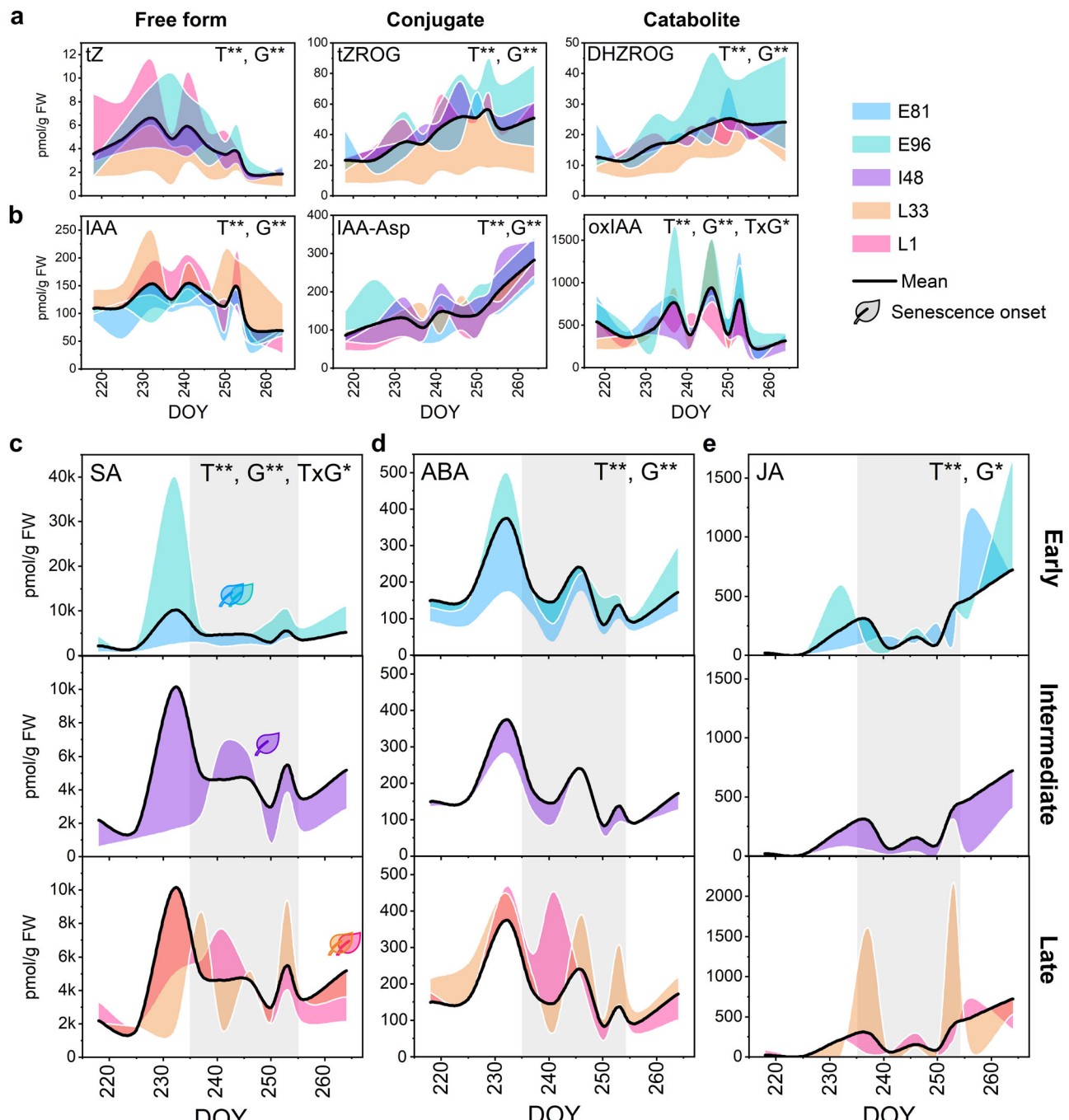

**Fig. 4 | Aspen genotypes show different dynamics of salicylic acid levels during autumn.** The levels of *trans*-zeatin (cytokinin, CK) metabolites (**a**), auxin (IAA) metabolites (**b**), salicylic acid (SA, **c**), abscisic acid (ABA, **d**) and jasmonic acid (JA, **e**) in the leaves of five SwAsp genotypes in autumn 2018 (all detected hormones in Fig. S8). The grey area represents the main stress period during autumn presented by the transcriptional responses (cascade-2 Fig. 3). The effects of time (T, nine time points 218–264 DOY), genotype (G) and their interaction (T×G) were tested with two-way ANOVA (FDR adjusted *P*-value < 0.01**, <0.05*). Metabolite levels are expressed relative to the mean, coloured area presents the difference between the genotype mean and the overall mean (*n* = 2–3) in each time point per genotype, see the details of n and statistical test results in Supplementary Data 14. Source data are provided as Source Data files. tZ *trans*-zeatin, tZROG *trans*-zeatin-O-glucoside riboside, DHZROG dehydrozeatin-O-glucoside riboside, oxIAA 2-oxindole-3-acetic acid, IAA-Asp Indole-3-acetic acid-aspartate.

## Discussion

There must be a trigger for the spectacular phenomenon of autumn senescence, but what is it? This evidence, together with our earlier results have demonstrated the remarkable complexity of how aspen leaves can enter senescence, including within the same individual tree[12,14,15,27]. It is known that many internal and external factors can induce senescence, and that abiotic stress, immunity responses, cell death and senescence share many components in their signalling

network[28]. Those processes are also affected by temperature, light and humidity[29–31] that all change in parallel in autumn. Given our aim to understand how the trees decide when to senesce in natural conditions, we identified two interlinked regulatory cascades that followed either gradual or short-term changes in environmental conditions in autumn, both advancing senescence through enhanced SA catabolism and de-regulation of the SA signalling pathway that compromise defence responses and lead to senescence initiation. Basically, this

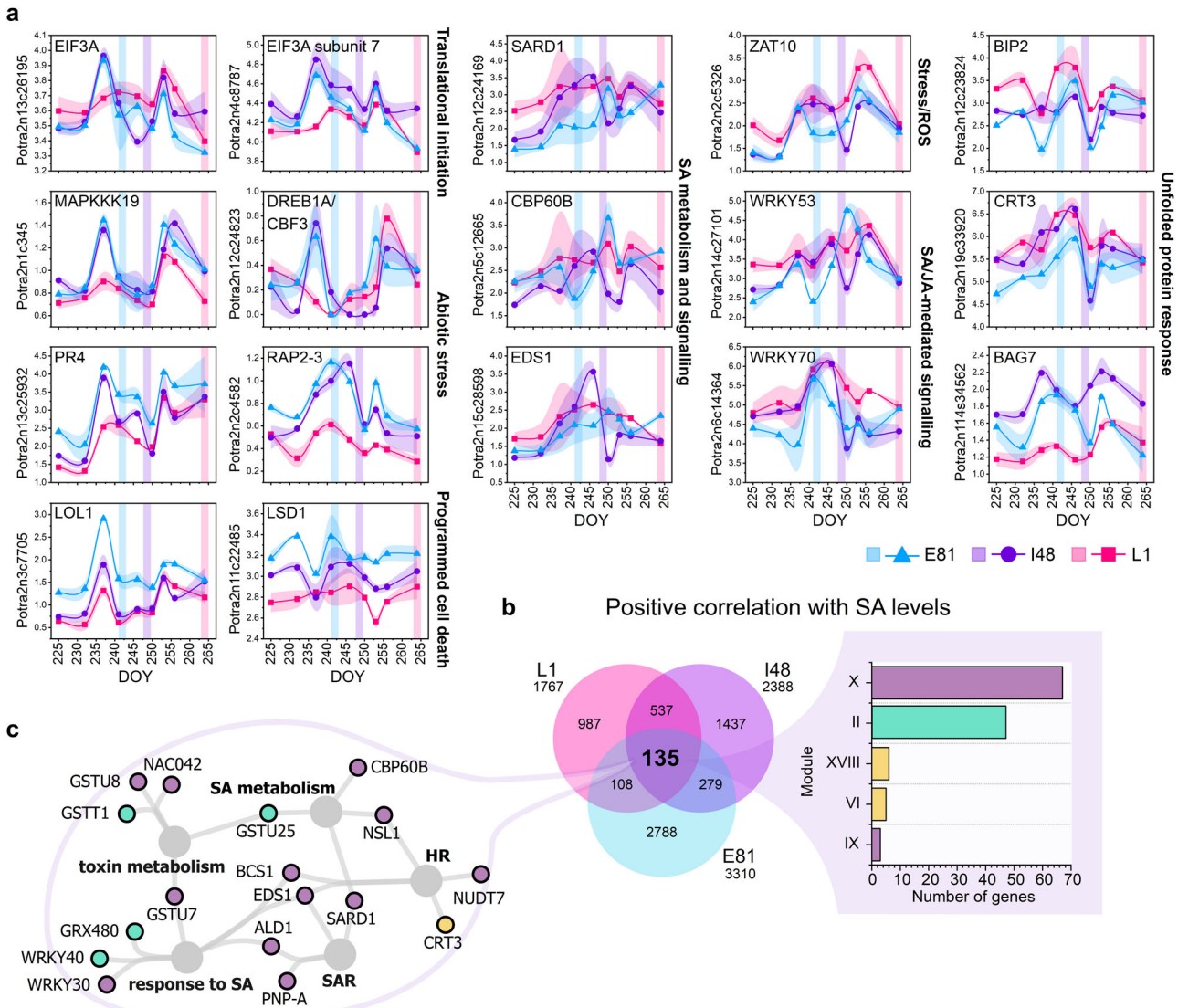

**Fig. 5 | Endogenous levels of salicylic acid affect the transcriptional response in aspen leaves.** The expression patterns of selected genes involved in translational initiation, abiotic stress responses, programmed cell death (PCD), the salicylic acid (SA)-mediated signalling pathway and unfolded protein response (UPR) (**a**). Data are mean ± SE (shadowed area), $n = 3$ in each time point per genotype, except $n = 2$ in I48 on 237 DOY. A shaded vertical line represents senescence onset in each genotype. Venn diagram displays the number of genes with significant positive correlation with SA levels in the three aspen genotypes (**b**). Bar chart shows the assigned modules and cascades for the genes with significant correlation (Pearson

$r$, $P$-value < 0.05 two-sided) with SA levels in three genotypes (**b**). Gene ontology (GO) term network was constructed with genes with positive correlation with SA levels in three genotypes (**c**). Significantly enriched GO terms are shown, and the nodes (genes) are coloured based on the cascade the genes were assigned to. See the expression patterns of additional genes related to the biological processes in Fig. S16, extended GO term network in Fig. S18 and the list of genes with significant correlation with SA in aspen genotypes in Supplementary Data 15. Source data are provided as Source Data files.

means that trees do not necessarily rely on only one pathway and environmental signal to properly time their autumn senescence. Any of the challenging environmental conditions in autumn such as the decrease in air temperature and changes in light conditions can escalate the metabolic constraints and pro-senescence signals that will open "the gateway to senescence" creating a situation where *"all roads lead to Rome"*. Our study demonstrates that even if this gate is opened, the passage through can be delayed to a certain extent by sustained defence mechanisms mediated largely by SA.

Studies in various plant species have led to the identification of thousands of genes that are induced or repressed during autumn, developmental or stress-induced senescence[21]. Our comparison of aspen and poplar transcriptome data[21,22] showed that the most conserved responses in *Populus* leaves during autumn were the repression of chloroplast processes, and the induction of stress and defence

responses mediated by NAC and WRKY TF families associated with senescence across plant species. As shown here in *Populus* spp., NAC100 and WRKY75 are also regarded as important regulatory TFs induced in senescing leaves also in other deciduous tree species such as *Ginkgo biloba*[32] and *Liquidambar formosana* Hance (formosan gum)[33], suggesting conservation of regulatory mechanisms across tree species. Our results also suggest that there may be a link between chromatin remodelling and SAG expression in autumn. Indeed, epigenetic mechanisms such as histone modification and DNA methylation have been associated with dormancy regulation in trees[34], with the regulation of plant immunity and the SA pathway[35,36] and leaf senescence through SAG expression in Arabidopsis[37]. Whether epigenetic mechanisms play a role in the regulation of autumn senescence, for example by affecting the ability of trees to respond to external and internal cues, remains an intriguing topic for further studies.

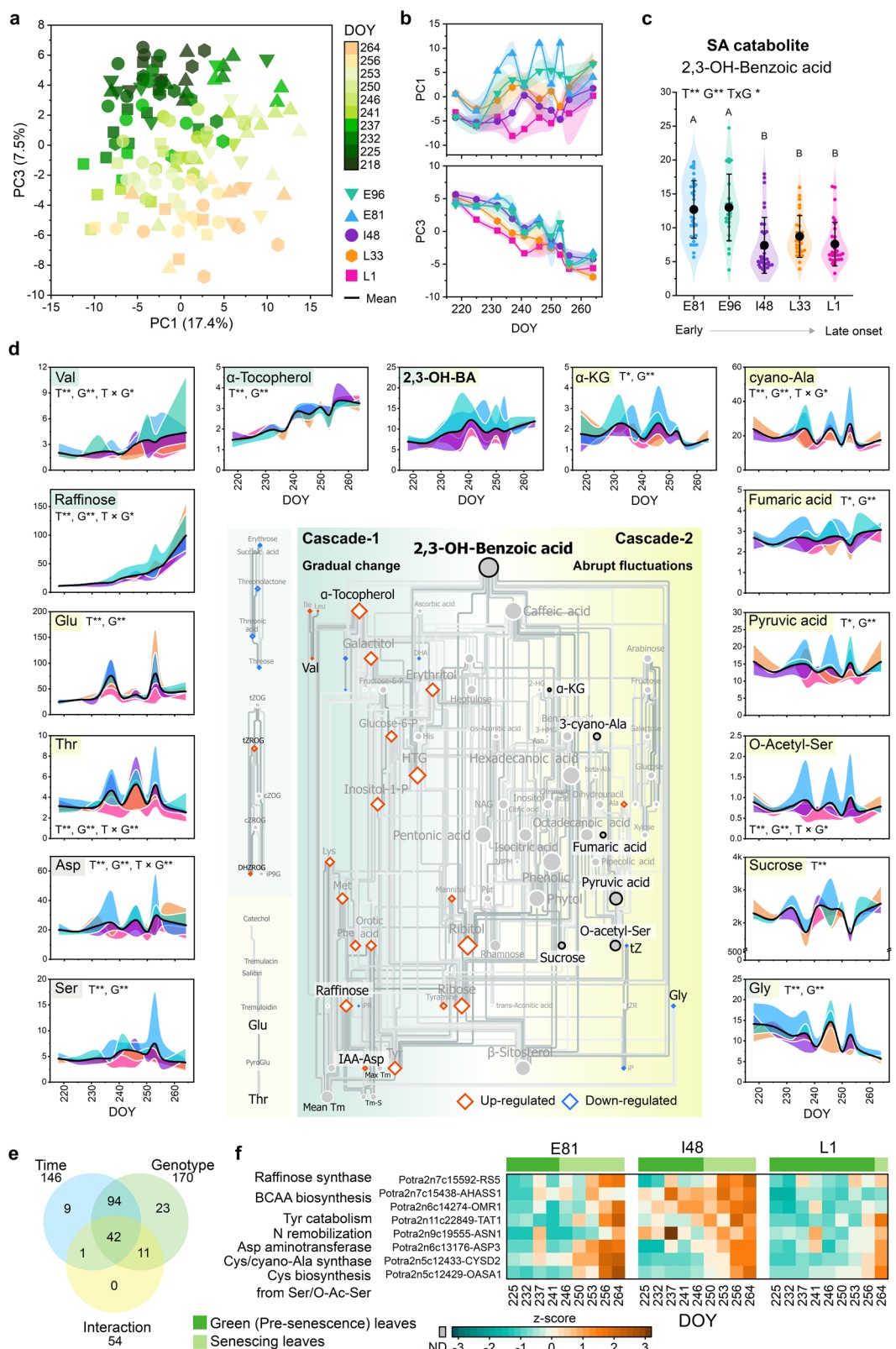

Since the genes with a gradual shift during autumn correlated with air temperature, cytokinin (CK) and auxin (IAA) levels, chlorophyll levels and metabolic senescence symptoms, they may modulate the rate of macromolecule degradation and advance senescence to ensure efficient nutrient resorption before frost kills the leaf cells, in line with the observation that cold temperature accelerates the rate of leaf senescence once it has been initiated in aspen[10]. Milder temperatures could give a slower senescence progression in early autumn, and more rapid progression in late autumn when the expression of SAGs and the levels of positive (JA) and negative hormonal regulators (CKs and IAA), have already changed; in other words, senescence is an active process, and the leaves respond to environmental factors after it has been induced. We propose that other factors beyond the gradual shift in the transcriptome and decrease in temperature, and CK and IAA

**Fig. 6 | Network analysis displays a connection between salicylic acid catabolite levels and metabolic responses during autumn.** Principal component analysis (PCA) including 183 metabolic markers in the leaves of five SwAsp genotypes in autumn 2018 (**a**). Time-dependent patterns of the first (PC1 17.4%) and the third principal components (PC3 7.5%) displaying variation across time points (**b**, see PC1 vs. PC2 in Fig. S20). Data are mean ± SE (shadowed area), n = 2–3 in each time point per genotype, see the details for n in a particular genotype and time point are in Source Data files and Supplementary Data 14. Genotypic levels of salicylic acid (SA) catabolite: 2,3-dihydroxybenzoic acid (2,3-OH-BA), during autumn (**c**). Violin plot shows the data points (n = 25 in E96, n = 26 in L33, n = 29 in E81 and L1, and n = 30 in I48), black dot presents the mean, and the whiskers SD (**c**). Temporal patterns of selected metabolite levels and the metabolic network structure visualised with the consistent metabolite relationships in all five genotypes in autumn 2018 (between 218–264 DOY, (**d**). Metabolite levels in SwAsp leaves are expressed relative to the mean, coloured area presents the difference between the genotype mean and the overall mean (**d**). The significant relationships (edges with P-value < 0.05 two-sided) are shown and coloured based on the correlation coefficient (darker colour depicts higher absolute Pearson r), the edges common for all five genotypes are shown (**d**). The size of the node is proportional to its degree, larger node size depicting higher connectivity. Diamond shapes mark up- and downregulated metabolites in senescing/late-senescence phase leaves compared to green leaves (Fig. S21, Supplementary Data 16). Venn diagram (**e**) depicts the number of metabolite markers with significant time (T, nine time points 218–264 DOY), genotype (G) and interaction (T × G) effects determined with two-way ANOVA (FDR adjusted P-value < 0.01**, <0.05*), Post-Hoc test for genotype were performed with Fisher's Least Significant Difference (LSD), different letters present significantly different means (P < 0.05 two-sided, details in Supplementary Data 14). Heatmap shows the expression of genes involved in the metabolic processes in three SwAsp genotypes (**f**). Values are mean VST-counts normalised to z-scores. Source data are provided as Source Data files and Supplemental Data sets.

levels predominantly coordinate the start of the senescence process in aspen.

Many of our observations connect senescence with abiotic stress responses, and an overlap in the regulation of plant immunity and the senescence network has been noted[7,38]. Stress-induced hormones and stress-responsive genes are typically regarded to promote senescence and considered as SAGs since their levels and expression are elevated during stress-induced and developmental senescence in many plant species[5,6,39,40]. However, we could not explain senescence onset in aspen in natural conditions by the accumulation of any single hormone; instead, our study highlighted that the genotypic sensitivity to environmental cues and their ability to activate and maintain SA-mediated defence mechanisms and antagonistic phytohormone interactions under stress are crucial for modulating the timing of senescence onset in autumn.

Transcriptional and metabolic responses leading to senescence initiation were very similar among the aspen genotypes (Fig. 8). First, environmental variation induced genes encoding cytoplasmic translational initiation factors, and some of them have been associated with senescence regulation in Arabidopsis[41]. Furthermore, blocking 80S ribosomes (cytoplasm) with cycloheximide can delay senescence indicating that some proteins synthesised by them can be central for predisposing plants for senescence[42]. In our study, this early transcriptional response was accompanied by upregulation of AP2/ERF TFs that typically respond to ABA or ET, and RAV TFs that are involved in the regulation of ET signalling, senescence and the photoperiodic pathway[43,44]. At the same time there was a shift in the expression of PCD regulators and in SA-mediated signalling pathway that was first enhanced and later repressed coinciding with senescence onset. The SA pathway appeared to be regulated by *SARD1* and *CBP60B* that are known to promote SA biosynthesis and regulate signalling downstream of immune receptors to induce SAR or HR in Arabidopsis[45,46]. Our study revealed that like SA, also glycerol-3-P which is another mobile signal inducing SAR[25,26] may play important role in the regulation of the onset of deciduous senescence. The timing of the shift in the expression of pro-senescence relative to anti-senescence factors depended on the genotype predisposing the trees for senescence, after which the repression of the SA-mediated signalling pathway appeared to potentiate those signals, compromise defence mechanisms and initiate senescence process.

Evidently, there are multiple layers of defence involved in the regulation of senescence timing under natural conditions. The early-senescence phenotype showed strong perturbations in primary metabolite and $H_2O_2$ levels, enhanced lipid peroxidation and low metabolic •OH scavenging capacity, low abundance of phenolic compounds including SA, high levels of SA catabolite and minor or no induction of SA pathway genes and enzymatic antioxidant systems, all contributing to its early-senescence onset. Although intermediate-senescing genotypes induced pro-senescence factors

at the same time as the early-senescing genotype, the SA pathway and antioxidant systems were also induced alongside and maintained for ca. ten days until repressed, and senescence was initiated. Late-senescing genotypes, on the other hand, did not induce pro-senescence factors in response to environmental conditions in early- or mid-autumn, instead they showed enhanced SA levels (ABA, JA and phenolic compounds), antioxidant systems and expression of genes that can alleviate redox status and ER stress. Only after pro-senescence factors were induced and SA catabolism enhanced later in the autumn did the SA levels and the expression of the SA pathway genes decrease, senescence was initiated, and it progressed fast. By then SAG expression had changed, senescence-associated metabolic symptoms appeared and the levels of positive (JA) and negative hormonal regulators (CK, IAA) elevated and depleted, respectively, all promoting senescence. Therefore, our results support the understanding that enhanced SA levels and the SA signalling pathway can repress autumn senescence onset in aspen trees under natural conditions potentially by antagonising pro-senescence factors induced before senescence and by promoting ROS metabolism and defence against ER stress, and not advancing senescence as often observed in annuals[16,17].

Our study adds to the growing evidence that SA plays a dual role in senescence and PCD regulation[47]. We found a positive relationship between the SA pathway and ROS metabolism, glycosylation and the unfolded protein response (UPR) that alleviate ER stress or induce cell death if prolonged[48,49]. SA can either supress ROS production[50], or amplify ROS signals such as those observed during senescence[18]. It can delay senescence by alleviating ER stress by modulating glycosylation of proteins and metabolites[51–53] and BiP proteins (ER chaperones) can act as positive or negative cell death regulators depending on the activation of SA-mediated HR[54]. It is yet unclear how SA signalling activates the UPR in response to ER stress[55,56], but our data suggest that *WRKY70*, *EDS1* and *CRT3* connect the SA pathway and ER stress regulon in aspen leaves. In Arabidopsis, WRKY70 negatively regulates developmental senescence and acts at the convergence point of antagonistic phytohormone pathways (inducing SA and repressing JA)[57,58]. EDS1 is involved in ROS- and SA-dependent cell death[59]. Calreticulins (CRTs) are molecular chaperones that participate in protein folding and quality control of glycoproteins[60,61], many of which are receptor proteins translocated via the trans-Golgi network to the plasma membrane. CRT3 is linked to the maturation of receptor-like kinases (RLKs) that affect defence responses and cell death[62,63]. Many of our results point to the differences in the perception and transduction of signals via receptor proteins and downstream signalling cascades that could explain genotypic sensitivity to environmental cues and metabolic signals such as ROS, calcium and hormones, and account for senescence phenotypes. The expression of genes encoding membrane-bound receptors in the cell periphery and in ER displayed significantly different dynamics between aspen genotypes and were

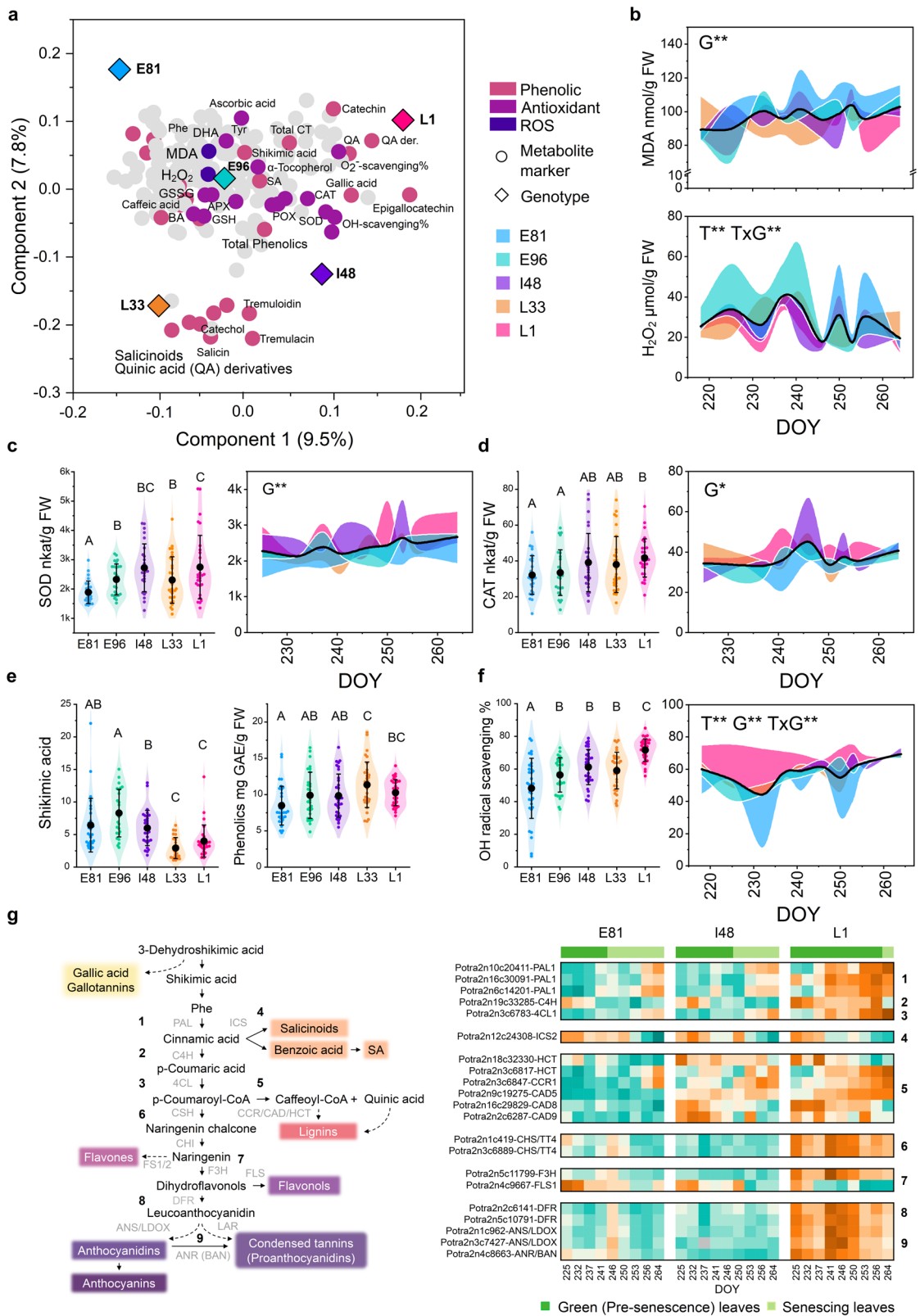

induced and repressed in response to environmental variation, and hormone levels, and potentially the receptor proteins failed to maturate due to prevailing ER stress in mid-autumn.

We found candidate markers associated with ROS metabolism that showed variation based on senescence phenotypes that we are also going to evaluate in the future under stress-free conditions. An obvious possibility is that the abrupt metabolic perturbations in

autumn occur due to disrupted redox status that can be evoked by a multitude of factors, with effects cascading throughout cellular metabolism. Indeed, in genotype I201, metabolic responses in mid-autumn were accompanied not only by temporarily increased lipid peroxidation and ROS levels but also by enhanced antioxidant systems. In general, the responses in ROS levels and antioxidant systems were transient, in line with results from other deciduous tree species

**Fig. 7 | Senescence phenotypes are related to the capacity of the genotype to induce and sustain enzymatic and metabolic antioxidant systems.** OPLS-DA (Orthogonal Projections to Latent Structures Discriminant Analysis) plot (the first two predictive components) shows the separation of five SwAsp genotypes based on the profile of 183 metabolite markers (**a**). ROS and metabolites with profound antioxidant functions are coloured (**a**). The temporal patterns of malondialdehyde (MDA nmol/g FW) and hydrogen peroxide ($H_2O_2$ μmol/g FW) levels (**b**). The temporal patterns and overall activities of superoxide dismutase (SOD, **c**) and catalase (CAT, **d**). The levels of shikimic acid (precursor) and total phenolics (**e**) and the hydroxy radical ($\cdot$OH) scavenging capacity (%) of the leaf extracts (**f**). Phenylpropanoid pathway and the expression of putative genes encoding enzymes in the pathway (**g**) in three SwAsp genotypes (z-score normalised values, mean $n = 2$–3 in

each time point per genotype). Levels and activities through the time course in each genotype (mean, $n = 3$) are presented relative to the overall mean, coloured area represents the difference between the genotype mean and the overall mean. Violin plots display all data points during the study period in each genotype (218–264 DOY, n = 25 in E96, $n = 26$ in L33, $n = 29$ in E81 and L1, and $n = 30$ in I48), black dot represents the mean and whiskers standard deviation (±SD). The effects of time, genotype and their interaction were tested with two-way ANOVA (FDR adjusted $P$-value < 0.05*, <0.01**). The different letters mark significantly different means between the genotypes (Fisher's Least Significant Difference LSD, $P$-value < 0.05 two-sided). See the details of statistical analyses in Supplementary Data 14 and other ROS markers in Fig. S23. Source data are provided as Source Data files and Supplemental Data sets.

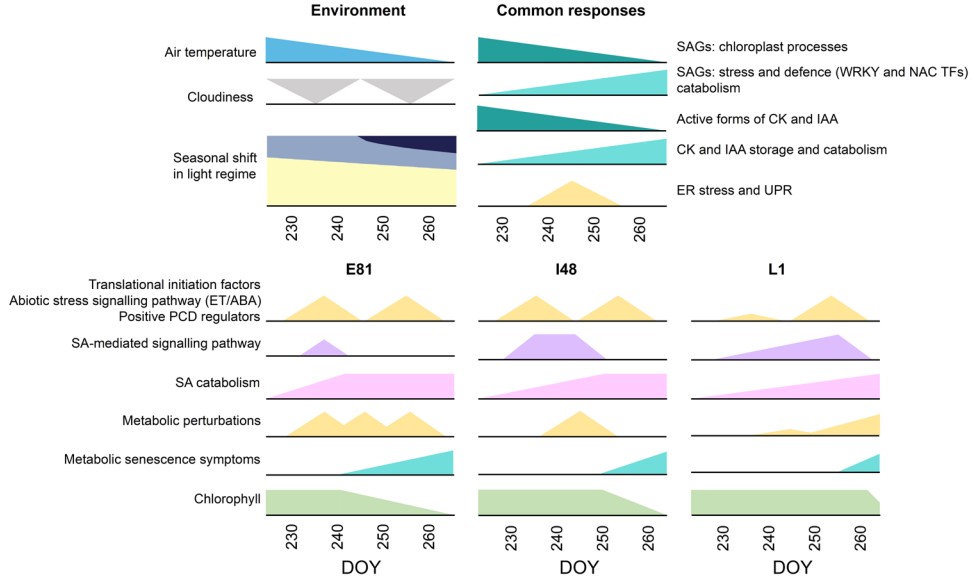

**Fig. 8 | Aspen genotypes show similar transcriptional and metabolic responses leading to senescence onset at different times during autumn.** Simplified illustration of environmental parameters and transcriptional and metabolic responses in aspen leaves in autumn 2018. Genotypes showed mainly similar changes in the expression of senescence-associated genes (SAGs) and in the levels of cytokinin (CK) and auxin (IAA) metabolites, that correlated with decreasing air temperature during autumn. In addition, the expression of genes related to endoplasmic reticulum (ER) stress and unfolded protein response (UPR) was enhanced in mid-autumn irrespective of genotype. At the transcriptional level, the

expression of genes involved in translational initiation, abiotic stress signalling in response to ethylene (ET) or abscisic acid (ABA) and programmed cell death (PCD) was enhanced, and genes related to salicylic acid (SA) metabolism and signalling pathway repressed at different times during autumn in aspen genotypes and those responses preceded and coincided with senescence onset, respectively. At the metabolite level, genotypes showed enhanced SA catabolism and perturbed primary metabolism before the onset. Typical metabolic senescence symptoms appeared around the same time as the chlorophyll content started to rapidly decline marking the initiation of nutrient recycling and senescence.

showing that redox homoeostasis is well maintained until advanced stages of leaf senescence[64].

Autumn is accompanied by ample variation in weather conditions and the light environment leading to many responses in aspen leaves. The changes in the leaf transcriptome, hormone and metabolite profiles that were evoked by these variations and represented by an intricate regulatory network support that trees integrate multiple environmental and internal signals to know when to senesce. In our earlier studies, we have simulated autumn conditions by shortening the photoperiod and by decreasing the air temperature[10,27]. Further studies are required to address how the changes in light spectral quality affect deciduous senescence and whether they could provide predominant seasonal cues for senescence. Furthermore, since senescence onset in the field conditions is linked with stress and SA signalling pathways, we continue to investigate whether the identified transduction pathways coordinate senescence timing in aspen also under stress-free conditions. Nevertheless, we would like to stress the importance of not only studying leaf senescence under controlled conditions, but also under field conditions to which plants are adapted, and where they are by default exposed to environmental fluctuations and multiple stress factors. The integration of multi-omics

techniques combined with the screening of physiological traits and genome wide association studies (GWAS) that we are currently performing in aspen populations has a great potential to generate novel hypotheses to decipher the complex regulation underlying senescence[65]. It may eventually enable the establishment of a diagnostic set of molecular markers for senescence in deciduous trees, which can differ from those in annual plants.

To conclude, we propose that after growth cessation and bud set, the decision to start senescing is not defined by one "master switch" but by two consecutive "switches"; first by the response of the trees to environmental factors that re-wire the regulatory network and induce and amplify pro-senescence factors and metabolic constraints. While this predisposes the trees for senescence, afterwards the timing of the initiation of the process itself—the start of the rapid chlorophyll depletion and the appearance of metabolic senescence symptoms that facilitate the catabolism of macromolecules and remobilisation of nutrients—is coordinated by SA metabolism and the SA signalling pathway and downstream post-transcriptional defence responses. Whereas the upregulation of SA levels and SA signalling can antagonise abiotic stress signalling and promote ROS metabolism and the UPR, their down-regulation compromises stress tolerance and potentiates

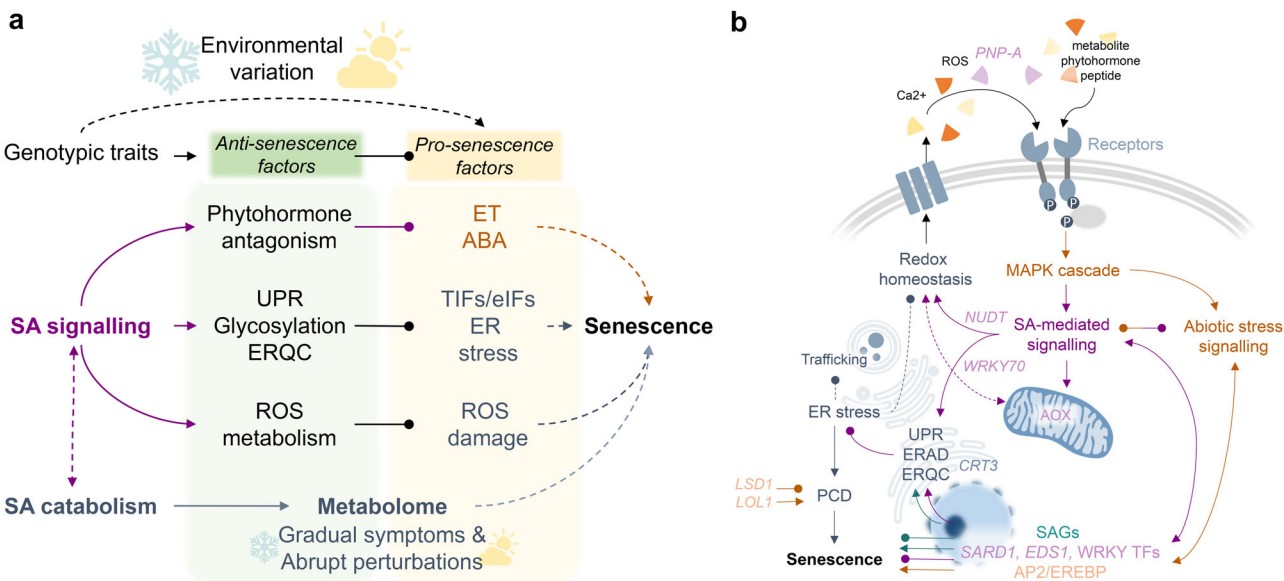

**Fig. 9 | Salicylic acid metabolism and signalling pathway coordinate the onset of autumn senescence in nature. a** Environmental variation leads to the re-wiring of transcriptional network and upregulation of genes encoding cytoplasmic translation initiation factors (TIFs/eIFs), ethylene (ET)-, abscisic acid (ABA)- and other abiotic stress-responsive genes along with positive programmed cell death (PCD) regulators. This response and other stress responses (ER stress, ROS formation) predominantly predispose the trees for senescence onset (pro-senescence factors). Gradually- and slowly-developing senescence symptoms and abrupt perturbations escalate the metabolic constraints resulting in senescence onset if counteractive defence responses that are largely mediated by salicylic acid (SA) signalling are not activated and maintained (anti-senescence factors).
**b** Hypothetical model displays the connections and potential mediators between biological processes involved in the regulation of autumn senescence onset. Genotypic sensitivity to external and internal cues is likely associated with receptor

proteins at the interface of signal perception. ER stress in autumn can hinder the maturation and establishment of receptors and thus signal transduction. The SA signalling pathway can promote protein glycosylation and protein folding via an ER-quality control (ERQC) system that alleviates ER stress. In addition, the SA signalling pathway can antagonise abiotic stress signalling activated before senescence onset and promote cellular respiration and redox status. Hence, we propose that autumn senescence onset in aspen is conditional upon the activation of pro-senescence factors in response to environmental conditions and evoked metabolic stress, whereas the timing of chlorophyll depletion can be delayed to a certain extent by upregulation of SA levels and the associated transcriptional programme that promote defence mechanisms. The de-regulation of SA metabolism and signalling thereby compromises cellular functions potentiating pro-senescence factors to initiate autumn senescence and developmental PCD in aspen leaves.

the pro-senescence signals initiating senescence. Thus, we propose that SA represses autumn senescence in nature, instead of promoting it as often reported in annuals in controlled conditions. The variation in senescence timing between aspen genotypes appears to be defined by how efficiently they can activate and sustain defence mechanisms under constantly challenging conditions in autumn to counteract the initiation of leaf senescence as a default programme in plant development.

## Methods
### Plant material
European aspen (*P. tremula*) genotypes selected for this study were genotype 201, a mature tree situated at Umeå University campus, and five other genotypes belonging to the Swedish aspen collection (SwAsp, planted 2004)[66]. Clonal replicates of 201 are a part of Umeå aspen collection (UmAsp, planted 2009–2010) and like the SwAsp trees they were grown in a common garden at the field station of Forest Research Institute (Skogforsk) in Sävar (63.9° N) near Umeå. SwAsp genotypes (1, 33, 48, 81 and 96) originating from different latitudes were selected for the study based on their senescence phenotypes. Each SwAsp genotype had at least three individual trees growing in the field, and they were 14 years old at the time of sampling in autumn 2018.

### Weather parameters and solar spectral quality
Weather data (air temperature, solar radiation, precipitation, air pressure, and relative humidity) for study years 2011 and 2018 were obtained from the Swedish Meteorological and Hydrological Institute (SMHI, http://shmi.se) and from the TFE weather station (http://www8.

tfe.umu.se/TFE-vader/, Department of Applied Physics and Electronics, Umeå University). Solar irradiance and spectral quality were measured with an ILT900-R Spectroradiometer (InternationalLight Technologies) on 14 dates during autumn 2020 between 240–268 DOY (the day of the year). Cloud cover data were based on an oktas scale, where 0 represents a clear sky and 8 is fully overcast (SHMI). Measurements were performed on sunny days with a clear or almost clear sky (scale 0-2, $n = 3$), partly cloudy days (scale 3–5, $n = 5$), and on almost fully overcast cloudy days (scale 6–8, $n = 5$) at Umeå University campus. Light spectra were measured 3–6 times around noon (12 a.m. to 1 p.m.). Each recorded measurement at 1 nm intervals was computed as an average of ten scans. Wavelength bands were assigned as blue (450–495 nm), red (620–700 nm) and far-red (700–750 nm). Those wavelength bands were used to calculate blue-to-red (B:R) and red-to-far-red (R:FR) ratios for sunny, partly cloudy and overcast days. Saturated vapour pressure (SVP) at given temperature ($T$, hourly mean) was calculated based on the equation: $SVP = (0.6108 \times exp[T \times 17.27/T + 237.3])$ and air vapour-pressure deficit (VPD) was derived from the equation: $VPD = SVP \times (1 - RH/100)$, where RH is hourly mean relative air humidity (%).

### Sampling
All aspen collections are grown in a common garden in Sävar, approx. 15 km distance from Umeå. Leaves were sampled twice a week from 6th of August (218 DOY) until 27th of September (270 DOY) in 2018. In genotype 96, leaf sampling was omitted on 270 DOY, since most of the leaves had abscised. Samples were collected from three individual trees per SwAsp genotype ($n = 3$). Sampling was always performed at noon and finished within one hour (12 a.m. to 1 p.m.). Five

healthy-looking short-shoot leaves were sampled from the middle level of the canopy (from four sides of the tree) using a telescopic tree pruner. The leaves were pooled, wrapped in aluminium foil and frozen immediately in liquid nitrogen. The sampling of leaves from genotype 201 situated at Umeå University campus in autumn 2011 is described in Edlund et al.[15].

## Chlorophyll content and the onset and rate of autumn senescence

Chlorophyll content index (CCI) of leaves (mean of five leaves) of the five SwAsp genotypes was determined twice a week from 6th of August (218 DOY) until 9th of October (282 DOY) in autumn 2018 with a chlorophyll metre (CCM 200 plus, Opti-Sciences). The CCI data of SwAsp genotypes in the field in 2011[11] and in the greenhouse study in 2006[10] and the data for genotype 201 in 2011[15] have been previously published. The CCI values (mean of five leaves) of its clonal replicates grown in the common garden were recorded in autumn 2018. In all cases, the senescence onset dates in autumn were determined based on the chlorophyll content index (CCI) as the day of the year (DOY) when rapid chlorophyll depletion started, using the curve fitting method[12] (OriginPro, Version 2021b. OriginLab Corporation, Northampton, MA, USA).

## RNA extraction and mRNA sequencing

The leaves (without petioles) were ground to fine powder in liquid nitrogen with a mortar and a pestle. Total RNA was extracted from 500 mg of leaf sample (fresh weight, FW) from three SwAsp genotypes (1, 48 and 81) from ten time points during autumn (225–270 DOY) 2018 using Spectrum™ Plant Total RNA isolation kit (Sigma-Aldrich) according to manufacturer's protocol ($n = 3$ individual trees per genotype and time point). DNA was removed with DNA-free™ DNA removal kit (Thermo Scientific). All the RNA samples were of good quality with OD $280/260 \geq 2.0$ assessed with a 2000 Nanodrop (NanoDrop Technologies, Wilmington, DE, USA) and RNA integrity number (RIN) $\geq 8.0$ determined with an Agilent 2100 BioAnalyzer (Agilent Technologies, Waldbronn, Germany). RNA concentration was determined with a Qubit fluorometer 2.0 using Qubit™ RNA BR Assay Kit (Thermo Scientific). mRNA sequencing was conducted by the Science for Life Laboratory in Stockholm, Sweden (SciLifeLab) using the Illumina NovaSeq 6000 platform. Raw data can be retrieved from the European Nucleotide Archive (https://www.ebi.ac.uk/ena/browser/home) under the accession PRJEB51801.

## Transcriptomic data processing

The data pre-processing steps were performed following the method based on Delhomme et al.[67]. The steps included quality control where the raw sequence data were first assessed (FastQC v0.10.1), and ribosomal RNA was removed (SortMeRNA v2.1b)[68] followed by trimming in order to remove adaptor sequences (Trimmomatic v0.32)[69] after which another quality control step was performed to ensure that technical artefacts were not introduced during the pre-processing steps. mRNA data were aligned with Salmon (v0.14.2)[70] using the *P. tremula* v2.2 cDNA library as a reference. Principal Component Analysis (PCA) and hierarchical clustering analysis (HCA) were applied in R (version 4.0.0) to assess the similarity among the biological replicates (biological quality control). One outlier sample had low library size and based on the PCA, samples from 270 DOY that were from yellowing leaves, were separated from the other samples. To reduce their contribution to the overall variation, they were excluded from the following analyses along with the outlier sample. For visualisation and downstream analyses, the counts were normalised using the variance stabilising transformation (VST) implemented in the Bioconductor DESeq2 package (v1.16.1)[71]. We have previously performed mRNA sequencing with genotype 201 in autumn 2011 (Gene Expression Omnibus [GEO] accession number GSE86960)[15] and here,

the RNA sequencing data were re-aligned with Salmon as described above to compare the data sets from the two study years. An overview of the mRNA data processing steps and parameters are available in the GitHub repository (https://doi.org/10.5281/zenodo.5906743)[72].

## Metabolomics

Phytohormone analyses and metabolite profiling were performed for the leaf samples of eleven time points of five SwAsp genotypes in autumn 2018 ($n = 2$–3 individual trees per time point and genotype) and of 12 time points of genotype 201 in autumn 2011. The levels of 23 phytohormones (pmol $g^{-1}$ FW) were quantified with UHPLC-ESI-MS/MS and metabolite profiling was performed with GC-MS as described in Supplementary Methods S1.

## Total soluble carbohydrates and starch assay

An aliquot (50 µL) of the leaf extract (soluble metabolite extract) was mixed with 200 µL of 5% phenol in water and 600 µL of sulfuric acid (97%). After incubation for 5 min at 95 °C, the samples were cooled down to room temperature, aliquots were transferred to 96-well plates and absorbance recorded at 490 nm with a microplate reader (SpectraMax 190, Molecular Devices). The absorbance of the blanks (contribution of the internal standards) and the absorbance at 600 nm were subtracted from the values. The concentration of total soluble carbohydrates (mg $g^{-1}$ FW) was calculated based on a standard curve of D-glucose.

Starch was determined with a protocol adapted from Smith and Zeeman[73]. After extraction of soluble metabolites, plant residues were dried using a speed vac and stored at −70 °C. Pellet was resuspended in 2 mL of $H_2O$, an aliquot (200 µL) was transferred to a new tube and heated at 100 °C for 15 min to gelatinise starch. Starch was digested to glucose with 200 µL of enzyme solution containing 6 U of α-amylase and 1 U of α-amyloglucosidase in 200 mM sodium acetate buffer (pH 5.5). Samples were incubated at 40 °C for 16 h after which they were heated at 100 °C for 2 min and cooled down to room temperature on ice. Samples were centrifuged for 5 min at 18,000 × $g$ at 4 °C. Glucose concentration was determined from the supernatant with glucose HK assay kit (Sigma). Assay volume was adjusted to 96-well plates and glucose concentration (mg $g^{-1}$ FW) was determined with a standard curve of D-glucose. Starch concentration (mg $g^{-1}$ FW) was calculated as an equivalent of anhydroglucose.

## Total phenolics and condensed tannins

Total content of phenolics was determined with Folin–Ciocalteu method[74] by measuring the absorbance at 765 nm and expressed as gallic acid equivalents (GAE mg $g^{-1}$ FW). Total content of condensed tannins (CTs) was determined with acid-butanol method as described by Porter et al.[75] based on a standard curve of procyanidin B2 (Sigma).

## Glutathione and signalling molecules

Leaf samples (~20 mg FW) were extracted with 1 mL of buffers/solvents and homogenised for 5 min in a bead mill with two tungsten beads and centrifuged for 15 min at 15,000 × $g$ at 4 °C. Glutathione levels (GSH, GSSG, total) were determined based on the 5,5-dithiobis(2-nitrobenzoic acid) (DTNB) glutathione recycling assay described by Salbitani et al.[76]. A thiobarbituric acid (TBA) test was used to determine MDA concentration (nmol $g^{-1}$ FW) as an indicator of lipid peroxidation[77]. Hydrogen peroxide ($H_2O_2$ µmol $g^{-1}$ FW) and methylglyoxal (MG µmol $g^{-1}$ FW) concentrations were determined by xylenol orange (FOX reagent) and DAB method, respectively[78].

## ROS-scavenging enzyme activities

Soluble proteins for enzyme activity assays were extracted from leaf samples (~50 mg) with 1 ml of cold 100 mM phosphate buffer (pH 7.8) containing 0.5 mM EDTA and 2% (w/v) polyvinylpyrrolidone-40, homogenised for 5 min in a bead mill with two tungsten beads, and

centrifuged for 15 min at 15,000 × g at 4 °C. Supernatants were kept on ice and enzymatic assays were performed immediately after the extraction. Catalase (CAT) and ascorbate peroxidase (APX) activities were determined using $H_2O_2$ and ascorbate, respectively, based on methods described by Aebi[79] with reaction volumes adjusted to 96-well microplates. Peroxidase activity (POX) was assessed using pyrogallol as substrate and based on the formation of purpurogallin[80]. Absorbance was recorded for 5–10 min, and the linear range was considered for calculating the rate of substrate depletion (APX, CAT) or product formation (POX). One unit of CAT was based on the extinction coefficient of $H_2O_2$ (43.6 M$^{-1}$ cm$^{-1}$ at 240 nm), APX of ascorbate (2.8 mM$^{-1}$ cm$^{-1}$ at 290 nm) and POX of purpurogallin (12.0 mM$^{-1}$ cm$^{-1}$ at 420 nm). Superoxide dismutase (SOD) activity was determined by a nitroblue tetrazolium (NBT) method described by Dhindsa et al.[81] with reaction volumes adjusted to 96-well microplate. One unit of SOD was considered to inhibit 50% of the photochemical reduction of nitroblue tetrazolium chloride (NBT). All assays included appropriate controls: dark controls (SOD), blank samples and reactions without the substrates. Enzyme activities (mean v) were normalised with fresh weight (nkat g$^{-1}$ FW) and with soluble protein content (nkat mg$^{-1}$ protein) of the samples determined by Quick start Bradford 1× dye (Bio-Rad)[82] using bovine serum albumin (BSA) as a standard. The results based on protein content and sample weight were comparable (Pearson r > 0.6, P-value < 0.001 two-sided). The enzyme activity levels based on sample weight are shown in the main manuscript.

## Superoxide anion radical and hydroxyl radical scavenging activity
Leaf samples (20 mg FW) were extracted with 1 mL of cold 70% MeOH following the same extraction protocol as for signalling molecules. Superoxide anion radical scavenging capacity of the leaf extracts was determined using PMS (phenazine methosulfate)-NADH system for superoxide generation[83]. Scavenging activity % was determined as the decrease in absorbance at 560 nm in the presence of leaf extract ($A_{sample}$) indicating scavenging of superoxide anions. Scavenging activity (%) was determined with the following equation: $[(A_{control} - A_{sample})/A_{control}] \times 100$, where $A_{control}$ is the absorbance without the leaf extract. Quercetin and superoxide dismutase were used as positive controls.

Hydroxyl radical scavenging capacity of the leaf extracts was determined based on the deoxyribose degradation method[84]. The reaction composition was as described in Li et al.[85]. Gallic acid and ascorbic acid were used as positive controls. Scavenging activity % was determined as described above based on the decrease in absorbance at 532 nm in the presence of leaf extract ($A_{sample}$) indicating inhibited degradation of deoxyribose due to hydroxyl radicals, i.e. hydroxyl radical scavenging capacity.

The spectrophotometric assays for secondary metabolites, glutathione, signalling molecules, ROS-scavenging enzyme and metabolic activities were adjusted for microplates (CAT and APX assays performed on UV-transparent 96-well plates, Cornic). The reactions that required a heating step (CTs, MDA, hydroxyl radical scavenging) were performed on 96-well PCR-plates that were closed during the incubation after which the aliquots were transferred on 96-well plates for absorbance measurements. Each sample was measured in duplicate, and the values of technical replicates were averaged prior to statistical analyses.

## Global gene expression and metabolite profiles during autumn
The overall variation in transcriptome and metabolome data in autumns 2018 and 2011 was visualised using principal component analysis (PCA, SIMCA P+, version 15, Umetrics, Umeå, Sweden). Data were log$_{10}$-transformed and scaled by unit variance. In addition to score plots, the changes in gene expression and metabolome were

visualised as time-dependent plots of PC scores. Subsequent Orthogonal Projections to Latent Structures Discriminant Analysis (OPLS-DA) was performed for metabolite data to identify markers accounting for the separation of the five SwAsp genotypes (SIMCA P+). The model was considered stable and reliable with four significant predictive components, R2 = 0.928 and Q2 = 0.886, cross-validation (CV)-ANOVA P-value < 0.001, and not overfitted based on permutation tests (100 times) that gave lower R2 and Q2 values for the permuted data sets than for the actual data set.

## Differentially expressed genes
Differentially expressed (DE) genes across time points and between genotypes in autumn 2018 were determined with the DESeq2 package, with false discovery rate (FDR) adjusted P-value < 0.01 considered significant in all cases. Time effect between specific time points was studied in two ways; by comparing time points against the first time point (225 DOY) and by comparing consecutive time points considering the whole data (all genotypes) set and within each genotype (log$_2$ fold change [FC] cut off >0.5)[86]. The difference in gene expression levels between individual genotype pairs was also tested. In addition, the main effects of time (between any time points), genotype (between any genotype) and their interaction effect (time × genotype i.e., different temporal patterns among the genotypes) were studied using likelihood ratio tests (LRT).

Senescence-associated genes (SAGs, up- and downregulated) were identified by comparing the first and the last time points during the period when the trees started to senesce (264–265 DOY). Since the genes showed considerable variation in their expression between consecutive time points (>0.5 log$_2$FC), gene lists were filtered based on cut-off >1.0 to find genes with a consistent increase or decrease over the autumn. The intersecting set of DE genes was visualised using Venn diagrams.

## Enrichment analyses
Gene Ontology (GO) term enrichment analyses were performed for identified gene sets with PlantGenIE using the aspen database (https://plantgenie.org)[87]. Since a greater proportion of Arabidopsis genes have annotation than do aspen genes, complementary GO term enrichment analyses and PFAM (protein family and domain) we performed using the best DIAMOND hits for Arabidopsis genes (PlantGenIE). Kyoto Encyclopedia of Genes and Genomes (KEGG) pathway enrichment analysis was performed with g:Profiler (https://biit.cs.ut.ee/gprofiler/gost)[88] and GO term networks were produced with ClueGO (version 2.5.7)[89] and CluePedia (version 1.5.7)[90] applications in Cytoscape (version 3.8.0) using the best DIAMOND hits for Arabidopsis gene IDs[91].

## Meta-analysis of senescence-associated genes in Populus spp. during autumn
A meta-analysis was performed to study the overlap between up- and downregulated genes during autumn in aspen (P. tremula) in 2011 (genotype 201) and 2018 (genotypes 1, 48 and 81) and in Populus spp. Genes that showed log$_2$FC > 1.0 between time points 217–265 (2011) and 225–264 DOY (2018) in aspen were regarded as senescence-associated genes (SAGs) either enhanced or repressed during autumn. The lists of consistent up- and downregulated genes in all aspen genotypes were compared with genes differentially expressed in poplar (P. trichocarpa) leaves during autumn[21,22]. The lists of up- and downregulated genes in poplar were obtained from the Leaf Senescence Database (LSD 3.0, https://bigd.big.ac.cn/lsd/poplar.php)[21] and from Lu et al.[22]. For comparative analyses, P. tremula gene IDs (Potra gene ID) were converted into poplar gene IDs (Potri gene ID) using the best DIAMOND hits obtained from PlantGenIE. The number of unique and common up- and downregulated genes during autumn between the two study years and between the two Populus species were visualised

using Venn diagrams. GO term enrichment and network analyses were performed for identified gene sets as described above.

## Statistical analyses of metabolomics data

Quality control (QC) samples that were included in each metabolite analysis and spectrophotometric assay batches were used to normalise the values to remove batch effects (MetaboAnalyst, version 5.0, https://www.metaboanalyst.ca/)[92]. In total, 183 individual metabolic markers were determined for five SwAsp genotypes, and 172 of them (except IAA, ABA and JA metabolites) were determined or had been determined earlier (CKs)[15] for genotype 201 in 2011. The main effects of time (T, the day of the year, DOY, across ten time points 218–264 DOY), genotype (G) and their interaction (T × G) effect on individual metabolite levels and total metabolite pools, enzyme activity levels, metabolic ROS-scavenging capacities and derived metabolite ratios were tested with two-way ANOVA (MetaboAnalyst). False discovery rate (FDR) adjustment was applied for ANOVA for multiple testing of individual metabolite levels. Post-hoc tests for genotype comparisons were performed with Fisher's Least Significant Difference (LSD). Correlation between senescence onset dates (from earliest to latest) and overall metabolic marker levels were studied with Spearman Rank correlation analysis.

## Senescence-associated metabolic markers

Statistical meta-analysis was applied to identify senescence-associated metabolic markers that were consistently up- or downregulated during autumn between green (pre-senescence) and senescing or late-senescence (269-270 DOY) leaves irrespective of SwAsp genotype in 2018 (MetaboAnalyst). The metabolic marker levels were scaled by unit variance, and FDR adjusted $P$-value and $\log_2$ fold change between the pre-senescence (first three time points, $n = 9$) and post-senescence samples (last two or three time points depending on the senescence phenotype, $n = 6$-$9$ in each genotype) were calculated for each genotype. Fischer's method was used to combine the $P$-values, and FDR adjusted $P < 0.05$ was considered significant. Next, we used pre-defined pre-senescence (44) and post-senescence (25) samples and identified the top biomarkers (5–15) for senescing leaves using Partial Least-Squares Discriminant Analysis (PLS-DA) models (Biomarker analysis, MetaboAnalyst). Since two out of the 15 top markers were not present in all the data sets, we applied the final model with 13 markers, and tested it for the classification of the rest of the samples from the SwAsp genotypes and from 201 into pre-senescence and senescing groups. The first time point when the sample was classified into senescing group based on metabolic markers was regarded as the senescence onset date in individual replicate trees. We then compared the senescence onset dates estimated by these markers and by the chlorophyll curve method. In addition to 2018 and 2011 data sets, we tested the model with another senescence experiment where the metabolite levels were determined for the leaves from control and girdled aspen stems[12].

## Network analyses

A weighted gene co-expression network analysis (WGCNA) was performed to identify gene modules and hub genes that are candidates to regulate biological processes in aspen leaves in autumn using the WGCNA package in R[23,24]. See details in the Supplementary Methods S2. In short, WGCNA was conducted with 21 602 genes that displayed temporal variation and were expressed in three SwAsp genotypes in 2018, and with 19,287 genes in genotype 201 in autumn 2011. Differential analysis was performed to study the preservation of the gene interconnectivity among the three genotypes in 2018. Hub genes with high intramodular connectivity were identified based on network topology structures. Enrichment analyses were performed for identified modules as described above.

To study the relationships between gene expression patterns and environmental factors and metabolic markers, weather parameters, chlorophyll content index (CCI), and metabolite levels were integrated in WGCNA and their correlation with eigengenes as well as individual genes in each of the three SwAsp genotypes and in the consensus network were interpreted. Gene and module co-expression networks were produced separately for the three SwAsp genotypes that were then merged (intersected) in Cytoscape to visualise only consistent relationships (edges present in three genotypes with FDR adjusted $P$-value < 0.05). Similarly, metabolite networks were produced separately for five SwAsp genotypes that were then merged (intersected) in Cytoscape to visualise only consistent relationships (edges present in five genotypes with FDR adjusted $P$-value < 0.05). The relationships between eigengenes and their correlations with weather parameters, chlorophyll levels and metabolic markers were confirmed with similar WGCNA analysis performed with genotype 201 in another year (2011). We considered the samples across 217–265 DOY in the network analyses because all the trees started to senesce within that period and since the focus was to find genes and metabolic markers that would account for the start of the process.

For clearer visualisation of gene co-expression network, edges with absolute correlation coefficient > 0.6 were included to focus on the connections between close neighbours of genes with significant positive correlations with SA levels. Nodes with high degree (connectivity) and high betweenness centrality are considered as important hubs and mediators, respectively, having a key regulatory role (hubs) or large control on the information flow through the network (mediators). The size of the node is proportional to its degree, larger size depicting higher degree. Edge colour intensity and thickness are proportional to the strength of the correlation, stronger colour and thicker lines representing higher absolute correlation (Pearson $r$), and when mentioned, blue and orange edges represent negative and positive correlations, respectively. Metabolite and module networks were visualised using a hierarchical layout and gene-gene and metabolite-gene networks using an organic layout.

## Reporting summary

Further information on research design is available in the Nature Portfolio Reporting Summary linked to this article.

## Data availability

Transcriptome data are available in the European Nucleotide Archive (ENA) under the accession number: PRJEB51801. Transcriptome data for 201 in 2011 are available in Gene Expression Omnibus (GEO) repository with accession number GSE86960. Other data generated in this study are available in the Supplementary data sets and Source data files provided with this paper. Poplar (*P. trichocarpa*) gene lists were obtained from the Leaf Senescence Database (LSD 3.0, https://bigd.big.ac.cn/lsd/poplar.php)[21] and from the Supplementary materials provided in Lu et al.[22]. Additional senescence data and chlorophyll levels in SwAsp genotypes were obtained from Supplementary Materials in Fracheboud et al.[10], Michelson et al.[11] and Edlund et al.[15].

## Code availability

Data processing steps and parameters for mRNA sequencing data are available in the GitHub repository (https://doi.org/10.5281/zenodo.5906743)[72].

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

## Acknowledgements

We thank the UPSC bioinformatics facility for their support in transcriptome data processing, Swedish Metabolomics Centre (SMC) for their support in metabolite analyses, Hans Stenlund (SMC) for providing the in-house SMC_RDA software and Skogforsk in Sävar for maintaining the SwAsp and UmAsp collections. This research has been supported by funding from the Swedish Research Council VR, Formas, Kempestiftelserna, Swedish Foundation for Strategic Research (SSF), Knut and Alice Wallenberg foundation, Vinnova, Trees for the Future (T4F) project and SE2B Horizon 2020.

## Author contributions

J.L., N.F., K.M.R. and S.J. planned the experiments, J.L., N.F. and K.M.R. collected the samples and performed the measurements in the field, J.L. and N.D. processed and analysed the transcriptomics data, J.L., J.Š. and O.N. performed the metabolite analyses, J.L. and N.F. performed the spectrophotometric assays, J.L. and P.B. analysed the physiological data, J.L. and S.J. wrote the paper and all authors contributed to the integration of the results and revision of the paper.

## Funding

## Competing interests

The authors declare no competing interests.
