## [Peer Review File · Nature Communications]

Reviewers' Comments:

Reviewer #1:

Remarks to the Author:

I was asked to specifically review the parts concerning the WGCNA analysis.

Briefly, the authors identified modules of co-expressed genes using a subset of the transcriptome and correlated the module eigengene and gene expression with environmental parameters, chlorophyll, and phytohormones. The methods are well described in the Supplementary Material file and seemed to be in line with the recommended procedures.

Concerning the methods, I would recommend being more explicit about how the genes used for WGCNA were identified. On page 3 of the SM, the authors state that genes with temporal variation and that were present in all genotypes (>50% of the samples) were used for WGCNA. It would help to clearly define how temporal variation was assessed (DE? At least one-time point?). It is also not clear to me what "present in all genotypes" and ">50%" mean.

I would also recommend making Fig 3 more accessible to the readers. The many panels (and panels within panels) and the rather short accompanying legend make it really hard to interpret. For example, it is not clear to me what is plotted in Fig 3A (leftmost panel). The legend states that I am looking at the "weighted co-expression network of eigengenes, weather parameter, and phytohormone levels.."; WGCNA is not of eigengenes nor of other parameters. Eigengenes are extracted from WGCNA-identified modules and can be correlated with other factors. I think I am looking at the eigengene expression, but it is not clear. I suggest rewriting the legend from scratch and possibly considering breaking the figure into multiple smaller ones.

Some additional minor revisions:

Line 95-96: I suggest rewriting what WGCNA does as "variation" and "gene relationships" do not really tell the reader how co-expression modules are obtained with this method.

Line 98: "relationship", I would be more specific and refer to correlation and its significant threshold.

Line 111-113: The role of epigenetic regulation is speculative and should be moved to the discussion.

Line 147: "explained this response". I would be careful in attributing any explanation to an observed correlation.

Reviewer #2:

Remarks to the Author:

Review of Nature Communications manuscript NCOMMS-22-45709

28 December 2022

I found this a fascinating study, whose results shed considerable new light on how autumn senescence is mediated in a perennial species, aspen. Crucially, the findings indicate that in this species salicylic acid, which in many annual species acts to promote senescence, actually represses it. It is the way genotype controls both the initiation of transcriptional/signalling responses to the changing light and temperature environment; and the point at which the repression of senescence by salicylic acid-mediated pathways is lifted; that determines the timing of senescence. There are, I think, important implications for our understanding of tree responses to their environment, especially in the context of conditions altering in response to climate change. The methodology appears sound to me (it capitalises on the invaluable SwAsp collection of aspens), and, importantly, the study was carried out in natural outdoor conditions. I do not have the expertise to comment on some of the statistical analyses. While the work has a focus on transcriptomics and bioinformatics, the results are supported by substantial biochemical analysis. The results are clearly presented and Figure 7A is a very nice summary of the model proposed by the authors for senescence regulation in aspen.

The paper is generally well-written and enjoyable to read, but the Materials and Methods section and figure legends will need some minor attention to the English, mainly to add definite/indefinite articles where appropriate. There is also a minor inconsistency in that 'colours' is spelt in the UK English style whereas 'defense' is US English usage.

Specific points, the last three being minor:

1. It would be very useful for the Results section to explain whether weather conditions in the two study years, 2011 and 2018, were similar and, if so, whether they were typical for the region. A Supplementary Figure or Table addressing this could be provided, based on the data described in the section 'Weather parameters and solar spectral quality' in Materials and Methods.
2. Line 56, 'senescence' should be 'senesce'.
3. Line 548, '14 000 rpm' - centrifugation conditions should be given in g, as in other parts of the Materials and Methods section.
4. Fig S3A - the legend should explain what the yellow leaf means. It is just labelled 'Senescence' - is this the first day on which leaf yellowing was visible?

Reviewer #3:

Remarks to the Author:

Comments to the authors

Leaf senescence is difficult to study, because it is intertwined with other phenological traits, not only in trees but in herbaceous plants as well. The experimental approach used in this work is very interesting, comparing aspen genotypes with different start dates for autumn leaf senescence. A very detailed analysis of hormones and metabolites was carried out, in addition to the analysis of changes in gene expression. One interesting result is that the transduction pathways and metabolic changes during leaf senescence are similar, and the difference between the genotypes is the time when senescence starts. This is why I think the first part of the title is not accurate. What it is shown is that the salicylic acid is important for leaf senescence in autumn, and this part should be stressed. The paper had answers about how senescence occurs, but not about why it starts earlier or late in the different genotypes. I think this is the important outcome of this paper, and should be stressed. I suggest to drop the first part of the title ("The timing of autumn") and left the rest.

Introduction

Lines 29 and 30: I do not think mature trees have more "stable" senescence patterns, I guess this comment refers to the fact that in younger trees the lower part of the canopy starts to senesce slightly earlier than the upper part. In my opinion, this is related to the fact that they are actively growing and producing new sylleptic branches with leaves that are younger than the ones produced in the lower, proleptic branches. But I do not think this implies an underlying difference in the senescence mechanism between young and mature trees.

Line 47. Citations of this could be added here.

Results

Lines 260-273: this result is not surprising, in order to salvage nutrients a working metabolic machinery is necessary, so a catastrophic increase in reactive oxygen species leading to PCD only could happen at the very end of the whole senescence process. The occurrence of PCD before is probably associated with fungal infections of the leaves.

Lines 286-288. It will be interesting, for future studies, to check if this mechanism is the same for plants growing without the stressful conditions that occur in the field. To be sure that the natural stresses that occur in the field are not inflating the role of salicylic acid and antioxidants systems in autumn leaf senescence.

Discussion

Regarding the text in general, I think a term like "Pro-death" should be avoided, and it will be better to use consistently "pro-senescence" and "anti-senescence" (instead of "pro-survival"). Autumn leaf senescence is basically a nutrient salvage process, deciduous plants discard their leaves in autumn and recycle nutrients to start to grow the following spring. The leaves die, but for a reason: to increase the probability of survival and the reproductive success of the tree.

Line 311 onwards: in addition to the comparison to *P. thichocarpa*, it will be interesting to add to

the Results/Discussion section a comparison to senescence in other tree species more distant phylogenetically from aspen, like ginkgo (<https://doi.org/10.1111/j.1399-3054.2004.00410.x>; doi/10.1073/pnas.1916548117) and Liquidambar (doi:10.1093/pcp/pcu160).

Line 361 onwards: The late senescing genotypes are from southern Sweden. They are growing far away from home, about 800-1000 km North. Maybe the higher antioxidant response is because they are stressed under these environmental conditions, compared with the autumn weather in the place of origin?

Line 406 onwards: for me it is not clear what is the signal for the start of autumn senescence the authors are looking for. Individual leaves can senesce at any time for different reasons (shading, biotic or abiotic stresses), the question is why all the leaves senesce and fall in autumn at the same time. I think it has to do with growth cessation and preparation for winter dormancy, but can be affected by other factors (nitrogen, stress). In consequence, the relationship between growth cessation and leaf senescence is not so straightforward as expected.

Line 415-415: maybe the increased ROS production was due to the occurrence of some stress (biotic or abiotic), that did not occur in the other experiment?

The evidence for a role of the SA in senescence presented in this paper is compelling. But as the authors explain, transduction pathways for senescence and stress overlap. As shown in a previous paper of the group (<https://doi.org/10.1104/pp.108.133249>), the leaves of these plants senesce after growth cessation, with no biotic or abiotic stress. I think the possibility of further studies under more controlled (or less stressed) conditions could be indicated in the discussion, in order to confirm that SA is important for senescence without the stress situation that usually occurs in the field.

Reviewer #4:

Remarks to the Author:

The authors have aimed to understand the cellular program leading to senescence onset in several genetically different aspen trees that vary substantially in their senescence onset dates in a common garden. Hence, they have integrated transcriptomics and metabolomics with co-expression network analyses to unveil why aspen genotypes start to senesce at different times, although grown in same location. The study revealed that the timing of autumn senescence initiation appeared to be controlled by two consecutive "switches" one is the environmental variation triggered by rewiring of the transcriptional network, stress signaling pathways and metabolic perturbations in a genotype-dependent manner and another one is the start of senescence process was defined by the ability of the genotype to activate and sustain stress tolerance mechanisms mediated by salicylic acid. The results revealed that salicylic acid represses autumnal leaf senescence onset in aspen in natural conditions by promoting defense mechanisms, rather than promoting it as often observed in annual plants. The experimental designing and execution are appreciable. The manuscript well framed and the theme of the study results are presented appropriately. However, some minor concerns need to be carried out in the manuscript. Therefore, I recommend the authors to incorporate the following changes to your revised manuscript.

Line 35: 'first in the genotype ... first and last in the genotype that senesces last' should be initially or primarily in the genotype / reframe the sentence.

Lines 40-42: Reframe it.

The manuscript needs a framework figure of listing all the analysis/steps done in this paper. Authors should check the formatting errors in throughout the manuscript. Rectify the issue throughout the manuscript.

Line 60: Fig. 1BC  Fig. 1B, C check the same throughout the MS.

Line 71: Concise the subtopics.

Line 84: Authors have mentioned 'several hundreds of genes were similarly up- or down-regulated during autumn irrespective of genotype' instead of several hundreds of genes should be mention the exact numbers of genes and their respective regulations. It will provide the clear knowledge to

the readers.

Line 89: 'defence responses encoding WRKY, NAC and TGA transcription factors (TFs)' it should be transcription factor families or mention the specific TFs like WRKY48, NAC055.

P- value 'P' should be in italics ('P'). Check throughout the manuscript including supplementary section and revise it.

If possible, the metabolites data can be represented through Mapman. If not, no issues.

Line 700: 'See details in the Supplementary Material' specify the S. material number for ease of reference.

Check the spelling of Signaling throughout the manuscript and follow the unique format.

Reviewer #5:

Remarks to the Author:

The manuscript describes an impressive amount of omics data and synthesis. Equally impressive is that it is grounded in a deep understanding of tree biology, physiology and gene regulatory networks, awareness of the advantages/limits of studies in both controlled and natural environments and also not confined to what seems to 'fit' with studies in Arabidopsis or other herbaceous plants. It presents a major advance in our understanding of (and way of thinking about) the mechanisms underpinning a process of global significance, leaf phenology.

Overall, the authors have done a good job of analyzing large complex data and presenting the analyses. A few suggestions to improve this:

While it makes sense that the role of SA metabolism and signaling is the focus of the summary figure 7, another summary figure is needed to help readers put "the pieces" together. In this case, summarize main changes (general patterns) in hormones, expression patterns (e.g., key modules related to pro-senescence and senescence phases), and metabolites with senescence (CCI), and major environmental trends over time (DOY). Show patterns of an intermediate genotype or perhaps if not too complex early, mid, and late genotype patterns could be stacked in separate panels.

I realize due to Journal limits on figures this may need to be in supplemental, but I think for papers with such complex data sets and analyses, viewing a summary of key analysis outputs first makes it far easier for readers to then look at the actual/detailed results and evaluate the support for the different parts of the summary.

I think many will be interested in up- and down-regulated genes shared among the *P. tremula* genotypes and among the *P. tremula* trees with the *P. trichocarpa* studies. Supplemental data 2 has lists of these shared gene sets, but I would like to see how they relate to the other analysis in the paper (e.g., what network module they are in).

Supplemental data 4 provides list of Hub genes and their module membership, but I did not see this information for network genes as well as any of the processed expression data.

NCOMMS-22-45709

Author response to the comments in blue

Line numbers refer to the revised manuscript file

REVIEWER COMMENTS

Reviewer #1 (Remarks to the Author):

I was asked to specifically review the parts concerning the WGCNA analysis. Briefly, the authors identified modules of co-expressed genes using a subset of the transcriptome and correlated the module eigengene and gene expression with environmental parameters, chlorophyll, and phytohormones. The methods are well described in the Supplementary Material file and seemed to be in line with the recommended procedures. Concerning the methods, I would recommend being more explicit about how the genes used for WGCNA were identified. On page 3 of the SM, the authors state that genes with temporal variation and that were present in all genotypes (>50% of the samples) were used for WGCNA. It would help to clearly define how temporal variation was assessed (DE? At least one-time point?). It is also not clear to me what “present in all genotypes” and “>50%” mean.

Author response:

Thank you for your constructive comments on our manuscript. We have now revised the data pre-processing steps of the WGCNA method (Supplementary Methods S2, page 4 of Supplementary Materials file).

We did not filter genes based on differential expression, since it is not generally recommended for WGCNA. First, we filtered out genes with low counts that were not expressed in at least 50% of all samples (i.e. in at least 38 out of 76 samples). Then, we filtered out genes that were not expressed in all genotypes, in other words, the gene had to be expressed in at least one time point in all of the three genotypes. Last, we filtered genes based on temporal variation and omitted genes with CV (coefficient of variation) of less than 0.1. At the end, the network was constructed with 21 602 genes which is 67% of the genes that were expressed in aspen leaves. Out of 37 075 genes 32 416 were expressed in at least one sample.

Supplementary Methods S1, Page 4: First, we filtered out low count genes that were not expressed in at least 50 % of the samples (in 38 out of 76 samples). Next, genes were filtered out if they were not expressed in all three genotypes in at least in one time point. Last, genes that showed low variation in their expression over the time course with coefficient of variation of less than 0.1 were filtered out. At the end, the WGCNA was performed with 21602 genes.

I would also recommend making Fig 3 more accessible to the readers. The many panels (and panels within panels) and the rather short accompanying legend make it really hard to interpret. For example, it is not clear to me what is plotted in Fig 3A (leftmost panel). The legend states that I am looking at the “weighted co-expression network of eigengenes, weather parameter, and phytohormone levels.”; WGCNA is not of eigengenes nor of other parameters. Eigengenes are extracted from WGCNA-identified modules and can be correlated with other factors. I think I am looking at the eigengene expression, but it is not clear. I suggest

rewriting the legend from scratch and possibly considering breaking the figure into multiple smaller ones.

Author response:

Thank you for these suggestions. Indeed the previous version of figure 3 was rich in data and the caption relatively short. This was mainly due to the restrictions of the word limit of the figure captions in the main manuscript. Therefore, the data that was presented in figure 3 is now split in two figures (new numbers Fig. 3 and Fig. 5) and weather parameters are now shown in Fig. 1 (Fig. 1d, e, f). This allows longer captions to explain the presented data. We hope that these changes are adequate to make the figures easier to understand.

Fig. 3. Network analysis reveals two interlinked transcriptional regulatory cascades that respond to environmental cues and hormonal signals during autumn.

Weighted gene co-expression network analysis (WGCNA) was performed with transcriptome data from three aspen genotypes during autumn 2018. Network visualisation is based on the

correlation between eigengenes (the signature expression pattern of the identified gene modules), weather parameters (past 24 h) and phytohormone levels (**a**) using a hierarchical layout in Cytoscape. Significant correlations (P -value <0.05 in all genotypes) are depicted by edges that are coloured based on the correlation coefficient, darker shades depicting higher absolute Pearson r (**a**). The size of each node is proportional to its degree, larger node size depicting higher connectivity. Enriched Gene Ontology (GO) terms and Kyoto Encyclopedia of Genes and Genomes (KEGG) pathways in the gene modules are shown (**a**, see details in Data S4). Cascade-1 comprises of modules that display gradual expression changes during autumn. Genes that are repressed or induced during autumn are regarded as senescence-associated genes (SAGs, see details in Fig. 2 and Data S1). Modules in Cascade-2 show transient expression changes (**a**). The two cascades are interlinked and connected with two gene modules (IX, X) enriched with genes involved in salicylic acid (SA) metabolism and signalling (**a**). The patterns of selected eigengenes in three SwAsp genotypes (**b**), $n=3$ in each time point per genotype, except $n=2$ at 237 DOY in genotype I48. A shaded vertical line represents senescence onset in each genotype that coincides with the transient down-regulation of the two SA signalling modules (**b**). Modules are numbered based on the descending number of assigned genes and coloured based on the assigned cascade (**a**, **b**). See details and all eigengene patterns in Fig. S10. ER=endoplasmic reticulum, SAGs=senescence-associated genes, tZ =*trans*-zeatin, iP =isopentenyl adenine, IAA=indole-3-acetic acid, IAA-Asp= Indole-3-acetic acid-aspartate

Fig. 5. Endogenous levels of salicylic acid affect the transcriptional response in aspen leaves. The expression patterns of selected genes involved in translational initiation, abiotic stress responses, programmed cell death (PCD), the salicylic acid (SA)-mediated signalling pathway and unfolded protein response (UPR) (a). A shaded vertical line represents senescence onset in each genotype. Venn diagram displays the number of genes with significant positive correlation with SA levels in three aspen genotypes (b). Bar chart shows the assigned modules and cascades for the genes with significant correlation with SA levels in the three genotypes (b). Gene ontology (GO) term network was constructed with genes with positive correlation with SA levels in three genotypes (c). Significantly enriched GO terms are shown, and the nodes (genes) are coloured based on the cascade the genes were assigned to. See the expression patterns of additional genes related to the biological processes in Fig. S16, extended GO term network in Fig. S18 and the list of genes with significant correlation with SA in aspen genotypes in Data S15.

Some additional minor revisions:

Line 95-96: I suggest rewriting what WGCNA does as “variation” and “gene relationships” do not really tell the reader how co-expression modules are obtained with this method. Line 98: “relationship”, I would be more specific and refer to correlation and its significant threshold.

Author response:

Thank you for these comments. We have now revised the text accordingly.

L98-104: Therefore, we performed weighted gene co-expression network analysis (WGCNA, details in Supplementary Materials)^{23,24} that considers correlation between genes in individual genotypes as well as in a consensus network (results in Figs. S4-S14 and Data S3-S6). With this approach, we could compare the transcriptome patterns among the genotypes in terms of gene modules and find significant correlations ($P < 0.05$) of the signature expression patterns of the modules (eigengenes) and individual genes with weather parameters and metabolite levels (see result in Figs. S4-S14 and Data S3-S6) to identify the sources underlying the transcriptome responses.

Line 111-113: The role of epigenetic regulation is speculative and should be moved to the discussion.

Author response:

We have revised the text accordingly and mentioned epigenetic regulation in the discussion: L326-333: Our results also suggest that there may be a link between chromatin remodelling and SAG expression in autumn. Indeed, epigenetic mechanisms such as histone modification and DNA methylation have been associated with dormancy regulation in trees³⁴, with the regulation of plant immunity and the SA pathway^{35,36} and leaf senescence through SAG expression in *Arabidopsis*³⁷. Whether epigenetic mechanisms play a role in the regulation of autumn senescence for example by affecting the ability of trees to respond to external and internal cues, remains an intriguing topic for further studies.

Line 147: “explained this response”. I would be careful in attributing any explanation to an observed correlation.

Author response:

We agree, the text has been revised accordingly.

Reviewer #2 (Remarks to the Author):

Review of Nature Communications manuscript NCOMMS-22-45709

28 December 2022

I found this a fascinating study, whose results shed considerable new light on how autumn senescence is mediated in a perennial species, aspen. Crucially, the findings indicate that in this species salicylic acid, which in many annual species acts to promote senescence, actually represses it. It is the way genotype controls both the initiation of transcriptional/signalling responses to the changing light and temperature environment; and the point at which the repression of senescence by salicylic acid-mediated pathways is lifted; that determines the timing of senescence. There are, I think, important implications for our understanding of tree responses to their environment, especially in the context of conditions altering in response to climate change.

The methodology appears sound to me (it capitalises on the invaluable SwAsp collection of aspens), and, importantly, the study was carried out in natural outdoor conditions. I do not have the expertise to comment on some of the statistical analyses. While the work has a focus on transcriptomics and bioinformatics, the results are supported by substantial biochemical analysis. The results are clearly presented and Figure 7A is a very nice summary of the model proposed by the authors for senescence regulation in aspen.

Author response:

Thank you, we greatly appreciate these positive remarks on our manuscript.

The paper is generally well-written and enjoyable to read, but the Materials and Methods section and figure legends will need some minor attention to the English, mainly to add definite/indefinite articles where appropriate. There is also a minor inconsistency in that 'colours' is spelt in the UK English style whereas 'defense' is US English usage.

Author response:

Thank you for these comments. We have now edited the language carefully. Some of the inconsistency in the language was brought by the language settings of the software that was used to produce the figures. We hope that we have managed to find and fix all the inconsistencies.

Specific points, the last three being minor:

1. It would be very useful for the Results section to explain whether weather conditions in the two study years, 2011 and 2018, were similar and, if so, whether they were typical for the region. A Supplementary Figure or Table addressing this could be provided, based on the data described in the section 'Weather parameters and solar spectral quality' in Materials and Methods.

Author response:

Thank you for these suggestions, in the new version of Fig. 1 we present the weather parameters for the study periods during the two years 2011 and 2018 side by side (Fig. 1d). In addition, we present a bar chart of monthly precipitation sum that clearly demonstrates the main difference between the weather conditions in the two study years (Fig. 1e). Other weather parameters can be found in Supplementary Figure S6 along with an additional

summary table (Table S1, page 14 of supplementary materials) for monthly means and sums (July-October) for sunshine, precipitation and air temperature for the study years in comparison with the 10-year average (2010-2020). Based on these data one can see that autumn 2018 was drier than 2011 and that weather conditions in late summer and autumn show considerable year-to-year variation in this region. The variation in precipitation and other weather parameters did not seem to affect senescence onset in aspen genotypes. The local campus tree (genotype 201 included in the study), in particular, has shown remarkably consistent year-to-year senescence timing (Fracheboud et al 2009).

Fig. 1. Autumn senescence phenotypes.

Swedish aspen (SwAsp) genotypes originating from the south (L1, L33), central (I48) and north (E81, E96) of Sweden were grown in a common garden in Sävar near Umeå (marked with a star, **a**). Genotype I201 is local to Umeå, situated at Umeå University campus, and its clones are part of the Umeå aspen collection grown in a common garden in Sävar (**a**). The onset and the rate of autumn senescence were determined based on the chlorophyll content index (CCI). Chlorophyll curves (CCI values as a mean of five leaves) are shown for the campus tree, I201, in autumn 2011 and for the three replicate trees of SwAsp genotypes in autumn 2018 (**b**). The variation in senescence onset date over the two study years is shown in a box plot representing the mean (solid line), 25% and 75% quartiles and minimum and maximum values (whiskers) (**c**). Points present data over two study years, data are from one parent tree (2011) and four clonal replicates in the UmAsp collection (2018) of genotype I201 (n=5), and from three to four replicates of the SwAsp genotypes in each study year (n=6-7). Minimum and maximum air temperature (C°), minimum relative air humidity (RH %), maximum vapour-pressure deficit (VPD, kPa) and maximum solar radiation (W m⁻²) in autumn 2011 and 2018 (**d**) in Umeå. Precipitation (monthly sum, mm) in autumn 2011 and 2018 (**e**) and the seasonal shift in the light environment in Umeå (**f**).

Table S1. Sunshine, precipitation and mean air temperature in July-October in 2011 and 2018 and over ten years (2010-2020) in Umeå. Values in red are above and values in blue are below the 10-year average.

		2011	2018	2010-2020			
Month				Mean	SD	Min	Max
Sum of sunshine hours^a	Jul	304.1	394.5	349.3	64.0	225.1	394.5
	Aug	178.7	248.4	213.6	49.3	172.0	291.1
	Sep	94.3	185.6	139.9	64.6	92.0	218.0
	Oct	137.8	126.8	132.3	7.8	48.4	161.7
Days of precipitation	Jul	9	5	7	3	2	19
	Aug	11	10	11	1	4	19
	Sep	20	16	18	3	5	20
	Oct	11	12	12	1	6	21
Sum of precipitation (mm)	Jul	63.0	68.2	47.1	35.4	4.5	139.9
	Aug	74.3	28.6	63.5	35.4	18.2	123.9
	Sep	116.6	44.1	67.5	27.1	11.2	116.6
	Oct	59.2	54.3	69.1	51.6	6.2	166.4
Mean air temperature (C°)	Jul	17.0	19.2	18.1	1.6	14.2	19.2
	Aug	14.5	15.1	14.8	0.4	13.3	15.2
	Sep	11.7	10.1	10.9	1.1	8.7	11.7
	Oct	4.9	3.3	4.1	1.1	2.4	5.7

^a global radiation over 120 W/m²

2. Line 56, 'senescence' should be 'senesce'.

Author response:

Thank you for noticing this misspelling, it has been corrected.

3. Line 548, '14 000 rpm' - centrifugation conditions should be given in g, as in other parts of the Materials and Methods section.

Author response:

We agree. We have revised the methods and provide xg instead of rpm.

4. Fig S3A - the legend should explain what the yellow leaf means. It is just labelled 'Senescence' - is this the first day on which leaf yellowing was visible?

Author response:

The yellow leaf marks the estimated date for senescence onset in each genotype determined as the start of rapid chlorophyll depletion. We have added an explanation of this to the legends of figures where it appears (in main manuscript and in the supplementary figures).

Reviewer #3 (Remarks to the Author):

Comments to the authors

Leaf senescence is difficult to study, because it is intertwined with other phenological traits, not only in trees but in herbaceous plants as well. The experimental approach used in this work is very interesting, comparing aspen genotypes with different start dates for autumn leaf senescence. A very detailed analysis of hormones and metabolites was carried out, in addition to the analysis of changes in gene expression. One interesting result is that the transduction pathways and metabolic changes during leaf senescence are similar, and the difference between the genotypes is the time when senescence starts. This is why I think the first part of the title is not accurate. What it is shown is that the salicylic acid is important for leaf senescence in autumn, and this part should be stressed. The paper had answers about how senescence occurs, but not about why it starts earlier or late in the different genotypes. I think this is the important outcome of this paper, and should be stressed. I suggest to drop the first part of the title ("The timing of autumn") and left the rest.

Author response:

Thank you for the constructive comments on our manuscript. We have revised our manuscript in an attempt to make the message clearer. We have also adapted the suggested title "Salicylic Acid Metabolism and Signalling Coordinate Senescence Initiation in Aspen in Nature".

As we expected, our data showed that the trees integrate many external and internal signals via a complex network to time their senescence. We think that this is one of the main messages of our manuscript. The other one is that the changes in salicylic acid metabolism and signalling are particularly important internal factors affecting senescence onset in aspen in nature, but in a different way than would be expected based on reports of annual plant senescence.

Here we demonstrate that the genotypes show similar transduction pathways that lead to senescence initiation but at different times during autumn. Indeed, we do not understand why these responses are induced earlier in some genotypes and later in others. Based on these data, we hypothesise that it may be due to their different sensitivity to the changes in the light environment that causes the stress signalling pathways to activate at different times. However, since we cannot separate different stress factors that co-occur in field conditions, further research in controlled conditions is required to test this hypothesis.

Introduction

Lines 29 and 30: I do not think mature trees have more “stable” senescence patterns, I guess this comment refers to the fact that in younger trees the lower part of the canopy starts to senesce slightly earlier than the upper part. In my opinion, this is related to the fact that they are actively growing and producing new sylleptic branches with leaves that are younger than the ones produced in the lower, proleptic branches. But I do not think this implies an underlying difference in the senescence mechanism between young and mature trees.

Author response:

Yes, what we wish to say is that deciduous senescence in mature aspen trees is represented by uniform yellowing of the whole canopy, whereas young trees often show progressive senescence starting from the bottom leaves. This difference in the senescence pattern is likely due to the different developmental age of the leaves in young trees. The majority of tree canopy is comprised of short shoot leaves that are of similar age since they flush at the same time in spring. We are currently investigating how the timing of deciduous senescence in mature trees compares with the timing of progressive senescence in their clonal saplings during the same year under the same natural conditions. With these studies we aim to establish guidelines for senescence phenotyping of young aspen plants and to study whether the senescence mechanisms are similar in both cases.

We have revised the text accordingly:

L26-31: On the other hand, our approach can have several benefits compared to experiments performed under controlled conditions, as the latter cannot separate a developmental programme from a timetable¹⁴ that mature trees – that cannot be easily grown under controlled conditions – have more uniform senescence of the canopy than young plants that typically show progressive senescence starting from the oldest bottom leaves, and that trees are likely to integrate multiple signals to know that it is autumn.

Line 47. Citations of this could be added here.

Author response:

Thank you for pointing this out. Citations have now been added.

Results

Lines 260-273: this result is not surprising, in order to salvage nutrients a working metabolic machinery is necessary, so a catastrophic increase in reactive oxygen species leading to PCD only could happen at the very end of the whole senescence process. The occurrence of PCD before is probably associated with fungal infections of the leaves.

Author response:

We agree, and as we have shown in the supplementary Fig. S22, the strong induction of ROS and MDA levels in the leaves of campus tree at mid-September was accompanied by a strong induction of ROS-scavenging antioxidant systems. We have revised the discussion about ROS as follows:

L419-427: We found candidate markers associated with ROS metabolism that showed variation based on senescence phenotypes that we are also going to evaluate in the future under stress-free conditions. An obvious possibility is that the abrupt metabolic perturbations in autumn occur due to disrupted redox status that can be evoked by a multitude of factors with effects cascading throughout cellular metabolism. Indeed, in genotype I201, metabolic responses in mid-autumn were accompanied not only by temporarily increased lipid peroxidation and ROS levels but also by enhanced antioxidant systems. In general, the responses in ROS levels and antioxidant systems were transient, in line with results from other deciduous tree species showing that redox homeostasis is well maintained until advanced stages of leaf senescence⁶⁴.

Lines 286-288. It will be interesting, for future studies, to check if this mechanism is the same for plants growing without the stressful conditions that occur in the field. To be sure that the natural stresses that occur in the field are not inflating the role of salicylic acid and antioxidant systems in autumn leaf senescence.

Author response:

Yes, we agree that the role of salicylic acid and antioxidant systems in senescence regulation may be enhanced in natural conditions in the presence of multiple biotic and abiotic stress factors. As we have shown in supplementary figure S1, senescence phenotypes (early, intermediate, late) persisted also in greenhouse conditions under natural light without stress factors (biotic factors, cold, drought, nutrient limitation). We are currently conducting senescence studies in the greenhouse with clonal saplings of several aspen genotypes and the data collection and biochemical analyses are ongoing. In the future, we aim to address the role of autumn signals and salicylic acid in senescence regulation in more detail under stressful and stress-free conditions. We have now briefly mentioned these future studies in the discussion.

L429-447: Autumn is accompanied by ample variation in weather conditions and the light environment leading to many responses in aspen leaves. The changes in the leaf transcriptome, hormone and metabolite profiles that were evoked by these variations and represented by an intricate regulatory network support our finding that trees integrate multiple environmental and internal signals to know when to senesce. In our earlier senescence studies, we have simulated autumn conditions by shortening the photoperiod and decreasing the air temperature^{10,27}. Further studies are required to address how the changing light spectral quality affects deciduous senescence and whether it could provide predominant consistent cues for senescence. Furthermore, since senescence onset in field conditions is linked with stress and SA signalling pathways, we continue to investigate whether the identified transduction pathways coordinate senescence timing in aspen trees also under stress-free conditions. Nevertheless, we would like to stress the importance of not only studying leaf senescence under controlled conditions, but also under field conditions to

which plants are adapted, and where they are by default exposed to environmental fluctuations and multiple stress factors. The integration of multi-omics techniques combined with the screening of physiological traits and genome wide association studies (GWAS) that we are currently performing in aspen populations has a great potential to generate novel hypotheses to decipher the complex regulation underlying senescence⁶⁵. It may eventually enable the establishment of a diagnostic set of molecular markers for senescence in deciduous trees, which can differ from those in annual plants.

Discussion

Regarding the text in general, I think a term like “Pro-death” should be avoided, and it will be better to use consistently “pro-senescence” and “anti-senescence” (instead of “pro-survival”). Autumn leaf senescence is basically a nutrient salvage process, deciduous plants discard their leaves in autumn and recycle nutrients to start to grow the following spring. The leaves die, but for a reason: to increase the probability of survival and the reproductive success of the tree.

Author response:

Thank you for this comment, we agree with the suggested terms and have used them in the current version of the manuscript.

Line 311 onwards: in addition to the comparison to *P. thichocarpa*, it will be interesting to add to the Results/Discussion section a comparison to senescence in other tree species more distant phylogenetically from aspen, like ginkgo (<https://doi.org/10.1111/j.1399-3054.2004.00410.x>; doi/10.1073/pnas.1916548117) and Liquidambar (doi:10.1093/pcp/pcu160).

Author response:

Thank you for this suggestion. To keep the text within the allowed word limit, we have added discussion with these citations:

L319-326: Our comparison of aspen and poplar transcriptome data^{21,22} showed that the most conserved responses in *Populus* leaves during autumn were the repression of chloroplast processes, and the induction of stress and defence responses mediated by NAC and WRKY TF families associated with senescence across plant species. As shown here in *Populus* spp., NAC100 and WRKY75 are also regarded as important regulatory TFs induced in senescing leaves in other deciduous tree species such as *Ginkgo biloba*³² and *Liquidambar formosana* Hance (formosan gum)³³, suggesting conservation of regulatory mechanisms across tree species.

L425-428: In general, the responses in ROS levels and antioxidant systems were transient, in line with results from other deciduous tree species showing that redox homeostasis is well maintained until advanced stages of leaf senescence⁶⁴.

Line 361 onwards: The late senescing genotypes are from southern Sweden. They are growing far away from home, about 800-1000 km North. Maybe the higher antioxidant response is because they are stressed under these environmental conditions, compared with the autumn weather in the place of origin?

Author response:

This is indeed one possibility. It will be interesting to investigate in the future whether the levels of ROS or antioxidants show variation based on the latitude of tree origin or senescence phenotype under stress-free conditions.

Line 406 onwards: for me it is not clear what is the signal for the start of autumn senescence the authors are looking for. Individual leaves can senesce at any time for different reasons (shading, biotic or abiotic stresses), the question is why all the leaves senesce and fall in autumn at the same time. I think it has to do with growth cessation and preparation for winter dormancy, but can be affected by other factors (nitrogen, stress). In consequence, the relationship between growth cessation and leaf senescence is not so straightforward as expected.

Author response:

We have been interested to know if there is a key environmental or internal signal that would trigger deciduous senescence in aspen. As we expected, and based on the evidence presented in this manuscript, the trees integrate both external and internal signals via an intricate regulatory network to properly time their senescence onset. We have now revised the discussion in an attempt to clarify this (See lines 429-436). An investigation of metabolic and transcriptional network structures during the senescence process induced by different factors such as darkness, drought, cold, nutrient deficiency or biotic stresses would shed more light into the transduction pathways that can lead to senescence initiation.

The consistent timing of senescence onset in aspen trees suggests that there could be a seasonal cue that predominantly promotes senescence. At this latitude, most of the aspen genotypes senesce around mid-September (250 DOY), soon after the start of astrological night period on 245 DOY. Therefore, light variation and its effects on circadian regulation are potential signals that the trees could sense in autumn. Our understanding on the effects of light quality variation on senescence is very limited compared to the effects of photoperiod and temperature. Therefore, in the future, we aim to study how the variation in light conditions affects deciduous senescence in aspen.

We fully agree with the reviewer that the relationship between growth cessation and senescence is anything but straightforward. In our field studies we have observed that growth cessation and bud set can occur prematurely during summer drought, but this does not lead to early senescence onset. So far, we have not found a correlation between the timing of bud set and senescence in the Swedish aspen population or in local Umeå aspen population. Clearly, there is yet much to learn about how deciduous senescence is regulated in trees in response to environmental and internal signals. One interesting hypothesis is that leaf senescence could be influenced by dormancy and associated chromatin silencing. This aspect we have also briefly mentioned in the manuscript.

L326-333: Our results also suggest that there may be a link between chromatin remodelling and SAG expression in autumn. Indeed, epigenetic mechanisms such as histone modification and DNA methylation have been associated with dormancy regulation in trees³⁴, with the regulation of plant immunity and the SA pathway^{35,36} and leaf senescence through SAG expression in Arabidopsis³⁷. Whether epigenetic mechanisms play a role in the regulation of

autumn senescence, for example by affecting the ability of trees to respond to external and internal cues, remains an intriguing topic for further studies.

Line 415-415: maybe the increased ROS production was due to the occurrence of some stress (biotic or abiotic), that did not occur in the other experiment?

Author response:

We agree that this is one possibility. We have now revised the discussion about ROS and antioxidants.

L419-427: We found candidate markers associated with ROS metabolism that showed variation based on senescence phenotypes that we are going to evaluate in the future also under stress-free conditions. An obvious possibility is that the abrupt metabolic perturbations in autumn occur due to disrupted redox status that can be evoked by a multitude of factors, with effects cascading throughout cellular metabolism. Indeed, in genotype I201, metabolic responses in mid-autumn were accompanied not only by temporarily increased lipid peroxidation and ROS levels but also by enhanced antioxidant systems. In general, the responses in ROS levels and antioxidant systems were transient, in line with results from other deciduous tree species showing that redox homeostasis is well maintained until advanced stages of leaf senescence⁶⁴.

The evidence for a role of the SA in senescence presented in this paper is compelling. But as the authors explain, transduction pathways for senescence and stress overlap. As shown in a previous paper of the group (<https://doi.org/10.1104/pp.108.133249>), the leaves of these plants senesce after growth cessation, with no biotic or abiotic stress. I think the possibility of further studies under more controlled (or less stressed) conditions could be indicated in the discussion, in order to confirm that SA is important for senescence without the stress situation that usually occurs in the field.

Author response:

Thank you for this suggestion. Indeed, further studies under stress-free conditions are in progress and we wish to address these issues in the future. We have now mentioned this also in the discussion:

L429-447: Autumn is accompanied by ample variation in weather conditions and the light environment leading to many responses in aspen leaves. The changes in the leaf transcriptome, hormone and metabolite profiles that were evoked by these variations and represented by an intricate regulatory network support our finding that trees integrate multiple environmental and internal signals to know when to senesce. In our earlier senescence studies, we have simulated autumn conditions by shortening the photoperiod and decreasing the air temperature^{10,27}. Further studies are required to address how the changing light spectral quality affects deciduous senescence and whether it could provide predominant consistent cues for senescence. Furthermore, since senescence onset in field conditions is linked with stress and SA signalling pathways, we continue to investigate whether the identified transduction pathways coordinate senescence timing in aspen trees also under stress-free conditions. Nevertheless, we would like to stress the importance of not only studying leaf senescence under controlled conditions, but also under field conditions to

which plants are adapted, and where they are by default exposed to environmental fluctuations and multiple stress factors. The integration of multi-omics techniques combined with the screening of physiological traits and genome wide association studies (GWAS) that we are currently performing in aspen populations has a great potential to generate novel hypotheses to decipher the complex regulation underlying senescence⁶⁵. It may eventually enable the establishment of a diagnostic set of molecular markers for senescence in deciduous trees, which can differ from those in annual plants.

Reviewer #4 (Remarks to the Author):

The authors have aimed to understand the cellular program leading to senescence onset in several genetically different aspen trees that vary substantially in their senescence onset dates in a common garden. Hence, they have integrated transcriptomics and metabolomics with co-expression network analyses to unveil why aspen genotypes start to senescence at different times, although grown in same location. The study revealed that the timing of autumn senescence initiation appeared to be controlled by two consecutive “switches” one is the environmental variation triggered by rewiring of the transcriptional network, stress signaling pathways and metabolic perturbations in a genotype-dependent manner and another one is the start of senescence process was defined by the ability of the genotype to activate and sustain stress tolerance mechanisms mediated by salicylic acid. The results revealed that salicylic acid represses autumnal leaf senescence onset in aspen in natural conditions by promoting defense mechanisms, rather than promoting it as often observed in annual plants. The experimental designing and execution are appreciable. The manuscript well framed and the theme of the study results are presented appropriately. However, some minor concerns need to be carried out in the manuscript. Therefore, I recommend the authors to incorporate the following changes to your revised manuscript.

Author response:

We are grateful for these positive remarks and constructive comments on our manuscript.

Line 35: ‘first in the genotype ... first and last in the genotype that senesces last’ should be initially or primarily in the genotype / reframe the sentence.

Author response:

We have revised the text as follows:

L35-39: Theoretically, the changes that truly regulate the onset of autumn senescence should precede or coincide with its onset, initially in the genotype that starts to senesce early and later in the genotype that senesces late in autumn, assuming that they utilize similar transcriptional, hormonal and metabolic programs to integrate the signals to initiate senescence.

Lines 40-42: Reframe it.

Author response:

We have revised the text.

L42-50: We demonstrate that the trees integrate many environmental and internal signals through an intricate regulatory network to properly time their senescence onset. The

information is transduced via – at least – two parallel cascades that are interlinked with the salicylic acid (SA) signalling pathway that is repressed at senescence onset along with decreased levels of SA and increased levels of its catabolite 2,3-dihydroxybenzoic acid. Our study demonstrates that aspen genotypes display similar transduction pathways to initiate senescence, but at different times during autumn and that SA represses autumnal leaf senescence onset in aspen in natural conditions by promoting defence mechanisms, rather than advancing it as often observed in annual species^{5,16–19}.

The manuscript needs a framework figure of listing all the analysis/steps done in this paper.

Author response:

We have now prepared a framework figure that can be found in the supplementary materials (Figure S1). It lists the analyses that has been conducted in this manuscript and the data that has been published earlier.

Figure S1. Overview of the analyses conducted in this manuscript and the sources of the previously published data integrated with the data analyses. **a)** Fracheboud et al.¹³, **b)** Edlund et al.¹⁴, **c)** Michelson et al.¹⁵, **d)** Li et al.¹¹, **e)** Lu et al.,¹², **f)** Lihavainen et al.¹⁶

Authors should check the formatting errors in throughout the manuscript. Rectify the issue throughout the manuscript.

Line 60: Fig. 1BC  Fig. 1B, C check the same throughout the MS.

Author response:

We have corrected the formatting errors based on the journal guidelines.

Line 71: Concise the subtopics.

Author response:

As suggested, we have revised some of the subheadings (old version in italics and the current version in bold):

Early, intermediate- and late-senescing genotypes were consistently different in the field and in the greenhouse

Senescence phenotypes are consistent in the field and in the greenhouse

A gradual shift in global transcriptome profile through autumn does not explain senescence onset

The shift in the global transcriptome profile does not reflect the variation in senescence onset

Line 84: Authors have mentioned 'several hundreds of genes were similarly up- or down-regulated during autumn irrespective of genotype' instead of several hundreds of genes should be mention the exact numbers of genes and their respective regulations. It will provide the clear knowledge to the readers.

Author response:

As suggested, we have now added the numbers:

L87-90: In line with the variation seen in PCA, 511 and 353 genes were similarly up- or down-regulated, respectively, during autumn irrespective of genotype, and over the two years (Fig. 2d), and those genes are typically regarded as senescence-associated genes (SAGs).

Line 89: 'defence responses encoding WRKY, NAC and TGA transcription factors (TFs)' it should be transcription factor families or mention the specific TFs like WRKY48, NAC055.

Author response:

We have now mentioned specific TFs:

L90-95: We compared our list of genes with those obtained from two transcriptomics studies of poplar (Fig. 2e) (*P. trichocarpa*)^{21,22} and found consistently up-regulated transcription factors (TFs) during autumn in both species related to innate immunity, stress and defence responses such as *WRKY75*, *WRKY48*, *NAC100*, *NAC072* and *TGA1* and the repressed genes enriched with gene ontology (GO) terms related to chloroplast processes (Fig. 2e, f, Data S2).

P- value 'P' should be in italics ('P'). Check throughout the manuscript including supplementary section and revise it.

Author response:

We hope that we have managed to find and correct these throughout the manuscript.

If possible, the metabolites data can be represented through Mapman. If not, no issues.

Author response:

We appreciate this suggestion. We found it challenging to visualise the patterns in five genotypes with a time-course in a clear manner. Below is an example of the metabolism overview in genotype E81 (218-264 DOY) produced with MapMan. Instead of line graphs, we decided to show the data with heatmaps of metabolite groups in each genotype (Fig. S18, see below an example of E81 data, Fig. S18a).

Mapman figure of metabolite responses in genotype E81 in autumn 2018.

Figure S18. Overview of metabolite responses in aspen leaves in autumn. Heatmaps display the mean metabolite levels normalized to z-scores in the leaves of five aspen genotypes, E81 (a), E96 (b), I48 (c), L33 (d) and L1 (e) in autumn 2018 (218-270 DOY), and in the leaves of genotype I201 (f) in autumn 2011 (217-269 DOY).

Line 700: ‘See details in the Supplementary Material’ specify the S. material number for ease of reference.

Author response:

The references to supplementary materials have been corrected and the methods are now referred as Supplementary Methods S1 and S2.

Check the spelling of Signaling throughout the manuscript and follow the unique format.

Author response:

We have checked the language throughout the manuscript, we hope that we have managed to find and fix all the inconsistencies.

Reviewer #5 (Remarks to the Author):

The manuscript describes an impressive amount of omics data and synthesis. Equally impressive is that it is grounded in a deep understanding of tree biology, physiology and gene regulatory networks, awareness of the advantages/limits of studies in both controlled and natural environments and also not confined to what seems to ‘fit’ with studies in Arabidopsis

or other herbaceous plants. It presents a major advance in our understanding of (and way of thinking about) the mechanisms underpinning a process of global significance, leaf phenology.

Overall, the authors have done a good job of analyzing large complex data and presenting the analyses. A few suggestions to improve this: While it makes sense that the role of SA metabolism and signaling is the focus of the summary figure 7, another summary figure is needed to help readers put “the pieces” together. In this case, summarize main changes (general patterns) in hormones, expression patterns (e.g., key modules related to pro-senescence and senescence phases), and metabolites with senescence (CCI), and major environmental trends over time (DOY). Show patterns of an intermediate genotype or perhaps if not too complex early, mid, and late genotype patterns could be stacked in separate panels.

I realize due to Journal limits on figures this may need to be in supplemental, but I think for papers with such complex data sets and analyses, viewing a summary of key analysis outputs first makes it far easier for readers to then look at the actual/detailed results and evaluate the support for the different parts of the summary.

Author response:

Thank you, we are extremely happy for this positive response to our manuscript. We appreciate the constructive comments, and hope that we have managed to incorporate the suggested changes adequately in this version of the manuscript.

As suggested, we have now prepared another figure for the main manuscript (Fig. 8). It is a simplified illustration that summarises both the common temporal patterns and the different patterns among the aspen genotypes. We have used one early (E81), one intermediate (I48) and one late (L1) as examples.

Fig. 8. Aspen genotypes show similar transcriptional and metabolic responses leading to senescence onset at different times during autumn.

Simplified illustration of environmental parameters and transcriptional and metabolic responses in aspen leaves in autumn 2018. Genotypes showed mainly similar changes in the expression of senescence-associated genes (SAGs) and in the levels of cytokinin (CK) and auxin (IAA) metabolites, that correlated with decreasing air temperature during autumn. In addition, the expression of genes related to endoplasmic reticulum (ER) stress and unfolded protein response (UPR) was enhanced in mid-autumn irrespective of genotype. At the transcriptional level, the expression of genes involved in translational initiation, abiotic stress signalling in response to ethylene (ET) or abscisic acid (ABA) and programmed cell death (PCD) was enhanced, and genes related to salicylic acid (SA) metabolism and signalling pathway repressed at different times during autumn in aspen genotypes and those responses preceded and coincided with senescence onset, respectively. At the metabolite level, genotypes showed enhanced SA catabolism and perturbed primary metabolism before the onset. Typical metabolic senescence symptoms appeared around the same time as the chlorophyll content started to rapidly decline marking the initiation of nutrient recycling and the senescence process.

I think many will be interested in up- and down-regulated genes shared among the *P. tremula* genotypes and among the *P. tremula* trees with the *P. trichocarpa* studies. Supplemental data 2 has lists of these shared gene sets, but I would like to see how they relate to the other analysis in the paper (e.g., what network module they are in).

Supplemental data 4 provides list of Hub genes and their module membership, but I did not see this information for network genes as well as any of the processed expression data.

Thank you for this suggestion. We have now added the information of module memberships for the listed SAG genes (Data S2). The information of module memberships for all network genes in both 2011 and 2018 data sets are provided in .txt files Data S3 and S5, respectively.

In addition, the correlations between gene expression and metabolite levels are provided in .txt files Data S7-S13 and Data S15. We believe that since our analyses identified previously known as well as unknown gene-metabolite interactions, further mining of the data may prove useful for the discovery of new gene functions.

Reviewers' Comments:

Reviewer #1:

Remarks to the Author:

The authors have addressed all my comments. I have no further concerns and am delighted to offer my full support for the publication of this fascinating and comprehensive study.

Dario Cantù

University of California Davis

Reviewer #2:

Remarks to the Author:

I am happy that the authors have addressed the points I raised in my initial review (the only minor issue is that in their covering letter they refer to Supplementary Figure S6 as containing weather data, whereas it is actually Figure S7 that contains these data).

Reviewer #3:

Remarks to the Author:

The authors have addressed all the questions raised in the first evaluation of the manuscript, and the article can be published.

Reviewer #4:

Remarks to the Author:

Authors have addressed all my queries in the revised manuscript. The manuscript was technically and scientifically sound. Therefore, I endorse this manuscript for publication in its current form.

Reviewer #5:

Remarks to the Author:

I have no further comments, the authors have done a thorough job of addressing all reviewer comments and have enhanced their manuscript.

NCOMMS-22-45709A

Author response to the comments in blue

REVIEWERS' COMMENTS

Reviewer #1 (Remarks to the Author):

The authors have addressed all my comments. I have no further concerns and am delighted to offer my full support for the publication of this fascinating and comprehensive study.

Dario Cantù
University of California Davis

Reviewer #2 (Remarks to the Author):

I am happy that the authors have addressed the points I raised in my initial review (the only minor issue is that in their covering letter they refer to Supplementary Figure S6 as containing weather data, whereas it is actually Figure S7 that contains these data).

Reviewer #3 (Remarks to the Author):

The authors have addressed all the questions raised in the first evaluation of the manuscript, and the article can be published.

Reviewer #4 (Remarks to the Author):

Authors have addressed all my queries in the revised manuscript. The manuscript was technically and scientifically sound. Therefore, I endorse this manuscript for publication in its current form.

Reviewer #5 (Remarks to the Author):

I have no further comments, the authors have done a thorough job of addressing all reviewer comments and have enhanced their manuscript.

Author response:

Thank you all for your constructive comments that helped us to improve our manuscript. We are delighted to hear that we made an adequate job of addressing them. Considering the comment from Reviewer #2 we have double checked that the references to each figure and table are correct in this version of the manuscript.